# Engineering of a chitin deacetylase to generate tailor-made chitosan polymers

**Martin Bonin**[1,2], **Antonia L. Irion**[1], **Anika Jürß**[1], **Sergi Pascual**[2], **Stefan Cord-Landwehr**[1], **Antoni Planas**[2], **Bruno M. Moerschbacher**[1]*

**1** Institute for Biology and Biotechnology of Plants, University of Münster, Münster, Germany, **2** Laboratory of Biochemistry, Institut Químic de Sarrià, University Ramon Llull, Barcelona, Spain

* moersch@uni-muenster.de

**Data Availability Statement:** All relevant data are within the paper and its Supporting Information files.

## Abstract

Chitin deacetylases (CDAs) emerge as a valuable tool to produce chitosans with a nonrandom distribution of *N*-acetylglucosamine (GlcNAc) and glucosamine (GlcN) units. We hypothesized before that CDAs tend to bind certain sequences within the substrate matching their subsite preferences for either GlcNAc or GlcN units. Thus, they deacetylate or *N*-acetylate their substrates at nonrandom positions. To understand the molecular basis of these preferences, we analyzed the binding site of a CDA from *Pestalotiopsis* sp. (PesCDA) using a detailed activity screening of a site-saturation mutagenesis library. In addition, molecular dynamics simulations were conducted to get an in-depth view of crucial interactions along the binding site. Besides elucidating the function of several amino acids, we were able to show that only 3 residues are responsible for the highly specific binding of PesCDA to oligomeric substrates. The preference to bind a GlcNAc unit at subsite −2 and −1 can mainly be attributed to N75 and H199, respectively. Whereas an exchange of N75 at subsite −2 eliminates enzyme activity, H199 can be substituted with tyrosine to increase the GlcN acceptance at subsite −1. This change in substrate preference not only increases enzyme activity on certain substrates and changes composition of oligomeric products but also significantly changes the pattern of acetylation (PA) when *N*-acetylating polyglucosamine. Consequently, we could clearly show how subsite preferences influence the PA of chitosans produced with CDAs.

## Introduction

Chitin and its derivative chitosan are polymers consisting of $\beta$-1,4-linked *N*-acetylglucosamine (GlcNAc, A) and glucosamine (GlcN, D) units in differing ratios. Chitin, with a high percentage of GlcNAc units up to 100%, forms strong crystalline fibers, and due to its strength, it is widely found in nature as a structural building block in the exoskeletons of arthropods and insects or the cell walls of fungi and algae [1,2]. Due to its wide abundance in nature, it is targeted by a variety of enzymes such as chitinases found in most organisms, e.g., to cleave the chitin of invading pathogenic fungi, weakening the pathogens and at the same time generating immunogenic chitin oligomers [3]. To disguise their chitin, some fungi are thought to have

**Funding:** This work was part of the EU programme "NanoBioEngineering of BioInspired BioPolymers (Nano3Bio)", which was financed by the European Union's Seventh Framework Programme under grant agreement no. 613931 (to M.B., S.P., S.CL., A.P., B.M.). A.P. was further funded from Ministry of Science and Innovation (MICINN), Spain under grant no. PID2019-104350RB-I00. The funders had no role in study design, data collection and analysis, decision to publish, or preparation of the manuscript.

**Competing interests:** The authors have declared that no competing interests exist.

**Abbreviations:** AngCDA, *Aspergillus niger* CDA; BFE, binding free energy; CBD, chitin binding domain; CDA, chitin deacetylase; CE4, carbohydrate esterase family 4; CMB, chitin magnetic bead; COS, chitooligosaccharide; DMF, dimethylformamide; DP, degree of polymerization; $F_A$, fraction of acetylation; GlcN, glucosamine; GlcNAc, N-acetylglucosamine; HPLC-MS, high pressure liquid chromatography–mass spectrometry; L, loop; MBP, maltose binding protein; MD, molecular dynamics; MMGBSA, molecular mechanics generalized Born surface area; MT, motif; PA, pattern of acetylation; paCOS, partially acetylated COS; RI, refractive index; RMSF, root mean square fluctuation; SSM, site-saturation mutagenesis; WT, wild-type.

evolved chitin deacetylases (CDAs), hydrolyzing the acetyl groups, thus lowering the fraction of acetylation ($F_A$) and yielding chitosans [4]. Chitin-to-chitosan conversion was shown to reduce immune defense reactions of host organisms [5], whereas the knock-out of CDAs was shown to increase survival of infected host organisms [6]. Beyond pathogenic fungi, CDAs are also found in chitin-degrading bacteria [7,8], and they are involved in chitin modulation in insects [9].

Chitosans also have a considerable potential to be used as multifunctional biopolymers in diverse application fields. In contrast to chitin, which is insoluble in all common solvents, chitosans are soluble in slightly acidic aqueous solutions due to their higher percentage of GlcN units [10]. Thus, they can be used more easily, e.g., as plant strengthening and antimicrobial agents in agriculture or antiseptical material for wound dressings in medicine [11–13]. Depending on the application, the degree of polymerization (DP), the $F_A$, and even the pattern of acetylation (PA) of the chitosan used need to be adjusted for optimum performance [14,15]. Both DP and $F_A$ can be controlled during the chemical production route of commercially available chitosans, but the PA remains close to a random distribution [16–18]. In contrast, chitosans produced using CDAs or via *N*-acetylation of polyglucosamines using CDAs in reverse have different, nonrandom PAs impacting the biodegradability and even bioactivity of these polymers [19]. However, this enzymatic approach to control the PA is limited by the number of CDAs characterized in detail so far, as the PA produced strongly depends on the enzyme used.

Theoretically, oligomers up to DP5 with all possible PAs can be produced using the CDAs known and characterized to date [20]. In order to produce defined PAs for larger oligomers and, even further, to control the PA of polymers, new CDAs are needed, or available CDAs need to be characterized in more detail. In addition to exploiting the natural diversity of CDAs found in different organisms, an alternative approach would be the targeted engineering of known CDAs, as it has been done for other chitin-modifying enzymes [21], to alter the PA of their products. Both approaches require a detailed understanding of the structure–function relationships in these enzymes, to identify crucial molecular differences between CDAs that are responsible for the different PAs they produce.

So far, several key residues have been described, mainly located in the 5 conserved motifs (MT1 to MT5) defined for carbohydrate esterase family 4 (CE4) enzymes [22], to which CDAs belong. These motifs include the metal binding triad consisting of an aspartate (MT1) and 2 histidines (MT2), and the catalytic aspartate (MT1) and histidine (MT5) with their supporting residues asparagine (MT3) and aspartate (MT4), respectively. Furthermore, a hydrophobic pocket, formed by 2 residues (MT3 and MT5 or MT4 and MT5), has been described, accommodating the acetate methyl group upon hydrolysis [22,23]. All of these residues are located at subsite 0, i.e., the region of the binding site that receives the GlcNAc unit to be deacetylated. Residues further along the binding site at negative subsites towards the nonreducing end as well as at positive subsites towards the reducing end of the substrate have been investigated less frequently. We assume, however, that the preferences for GlcNAc or GlcN units at these nonzero subsites deeply influence the PA of the products, as suggested earlier [14,24,25]. Therefore, amino acid residues that interact with the substrate at these subsites should be studied in more detail.

In addition to determining the 3D structure of the enzyme–substrate complex, knowledge-based enzyme mutagenesis has been used in previous studies to investigate the function of specific residues along the substrate binding site [5,22,26]. Although proven effective, this approach is rather time consuming and, thus, limited to a small number of residues. As an alternative to the knowledge-based approach, a random mutagenesis library was successfully screened by Pascual and Planas to identify the VcCDA mutein K275E/H127R that has a

significantly increased activity on chitin tetramers [27]. Besides these in vitro experiments, in silico studies are increasingly performed to analyze the binding site in more detail. These include homology modeling with substrate docking [5,28,29] and molecular dynamics (MD) simulations [26,30–32]. Despite great advancements in these fields, in silico predictions still need to be backed up by laboratory experiments, or vice versa, but they can help to understand experimental results.

In this study, we aimed to combine these in silico and in vitro approaches to investigate the substrate binding site of a previously described CDA from the plant-endophytic fungus *Pestalotiopsis* sp. [4]. We created a site-saturation mutagenesis (SSM) library of 27 residues along the binding site, which we screened according to Pascual and Planas [27]. In silico analyses then helped to narrow down the most interesting residues that were screened and analyzed in more detail. Finally, muteins, which showed different subsite preferences on oligomeric substrates, were used to *N*-acetylate polyglucosamine to evaluate the impact of these mutations on the PA generated on polymers.

## Materials and methods

### Homology modeling, substrate docking, and MD simulations

A PesCDA (UniProt ID: A0A1L3THR9) homology model was generated with SWISS-MODEL [33] using ClCDA (PDB ID: 2IW0 [34]) as a template, including its zinc ion in the active site. As these were not part of the template, the signal peptide and the chitin binding domain (CBD) were removed, resulting in a homology model covering residues 26 to 238. The model was evaluated based on literature data for important CDA residues such as the catalytic and metal binding residues. It shows a good similarity to the AlphaFold2 model available in the AlphaFold Protein Structure Database with an RMSD = 1.341 Å on all atoms and an RMSD = 0.782 Å on the 27 SSM library amino acids (see S2 Fig) [35,36]. Ligand generation and docking was performed as described previously [32]. To find 3 conformations of each ligand in binding mode [−3,0] (for nomenclature explanation, see sec. Results), the grid box size and position were adjusted to exclude subsites +1 and +2. MD simulations with 3 conformations for each ligand and binding mode and subsequent root mean square fluctuation (RMSF), hydrogen bond, and molecular mechanics generalized Born surface area (MMGBSA) analysis were carried out as described previously [32].

### Generating the site-saturation mutagenesis library

The PesCDA construct created previously [4], containing both an N- and C-terminal Strep-Tag II and the maltose binding protein (MBP) cloned into the pET22b plasmid (pET22b::NSt-MBP-CDA-CSt), was used as a template for cloning in this work. Here, the PesCDA CBD was replaced by the 2 CBDs from VcCDA (UniProt ID: Q9KSH6), for which the screening method was originally designed [27]. The resulting construct pET22b::NSt-MBP-PesCDA-VcCBD1+2-CSt (see S12 Fig for the complete sequence excluding the plasmid backbone) was cloned into *E. coli* Rosetta 2 (DE3) cells to express the protein. To verify the usability of this constructs, the protein was purified both via Strep-tag II affinity chromatography as well as chitin magnetic beads (CMBs) purification, before successfully testing its activity on A5.

The PesCDA together with the VcCDA CBD's section of the construct was codon optimized by GeneArt (Thermo Fisher Scientific). This section (PesCDA-VcCBD1+2-opt) was reintroduced into the plasmid, and the resulting fusion construct is referred to as the nonmutated PesCDA, termed "PesCDA[nm]" in this article. For the library, the codons of 27 amino acid positions (sec. SSM library preparation and generation) were individually mutated by GeneArt to generate a pool of SSM PCR products containing all 19 non-wild-type amino acid

substitutions for each position. Each pool of PCR products was cloned into the original plasmid, which was then transformed into *E. coli* BL21 (DE3) cells. For each position, 64 colonies were stored as a glycerol stock in a 96-well plate format and used for a PCR to amplify the region of interest. For the PCR, the KAPA2G Fast ReadyMix (no dye) (Roche) was used in a total volume of 5 µl. Excess dNTPs and primers were removed by adding 0.65 µl Antarctic Phosphatase Buffer, 0.1 µl Antarctic Phosphatase, and 0.03 µl Exonuclease I (NEB) and subsequent incubation at 37˚C and 80˚C both for 15 min. The PCR products were sequenced by GeneArt and *E. coli* clones carrying the plasmids with the desired mutations were replated as new glycerol stocks in 96-well plates. Each 96-well plate contained up to 76 library and 4 PesCDA$^{nm}$ colonies. In total, 43 muteins could not be identified during sequencing resulting in 470 muteins in the final PesCDA SSM library.

## Complete library screening with A4

The screening assay is based on the method developed by Pascual and Planas [27] and was conducted on the same Automated Liquid Handling Bravo Platform from Agilent Technologies. Before running the screening with the SSM library muteins, it was first conducted using the PesCDA$^{nm}$ in comparison to an empty vector control using different dilutions of CMBs (see S1 Fig). Based on these results, we decided to use a 1:10 dilution of CMBs assuming that they are saturated with enzymes to have an upper limit for enzyme concentration. Besides this adjustment, only minor changes have been made for this library screening. Instead of 25 µl, 75 µl of the supernatant after cell lysis were transferred to a new plate containing 10 µl 1:10 diluted CMBs, which were previously washed twice with water and 3 times with 50 mM TEA buffer at pH 8.5 and finally resuspended in 50 mM TEA buffer at pH 8.5. Similarly, after incubating the CMBs to allow protein binding, the beads were washed twice with 50 mM TEA buffer at pH 8.5. Finally, the reaction was performed with a final concentration of 0.5 mM A4 in 50 mM TEA buffer at pH 8.5 in a total volume of 50 µl. To quantify the amount of free primary amines, a fluorescamine-based assay was used. As a standard, 4 glucosamine concentrations (0 µM, 75 µM, 150 µM, and 250 µM) were used in triplicates. For the quantification, 70 µl of 50 mM TEA buffer at pH 8.5 were mixed with 30 µl of the reaction supernatant or the glucosamine standard in a black 96-well plate before adding 20 µl of 2 mg/ml fluorescamine in dimethylformamide (DMF). The mixture was incubated for 10 min at room temperature before 150 µl 1:1 DMF:H$_2$O were added. Using the FLx800 (BioTek Instruments), the samples were mixed for 15 s before the fluorescence was measured with 360 ± 20 nm excitation and 460 ± 20 nm emission.

All data were evaluated with an in-house script, which executed the following steps. First, a regression line is calculated for the glucosamine standard to calculate the amount of free primary amines in each well. The average activity of 4 PesCDA$^{nm}$ per plate was set to one, and the muteins' activities were adjusted accordingly. Finally, the mean and standard deviation from 4 independent measurements for each mutein were calculated. As no activity or very high activity was measured in some wells, these outliers were automatically removed with the following method: If the standard deviation is higher than 0.2, the value furthest away from the mean is eliminated. This elimination is withdrawn if it does not decrease the standard deviation by more than 20%. A detailed list of all values with and without the outlier elimination can be found in S1 Data.

## Detailed screening of 7 positions with A4

The muteins for 7 positions (N75, W76, Y140, F141, D164, Y168, and H199) were screened again by hand using A4 as a substrate, which was produced as described previously [32]. First,

1.5 ml LB$_{Amp100}$ autoinduction medium [37] in 96-deep well plates were inoculated with 10 μl from the glycerol stocks. The plates were sealed with breathable foil and incubated at 26˚C, 240 rpm. After 48 h, the plates were centrifuged at 4˚C, 2,500 rpm for 20 min. The supernatant was discarded and the pellets were stored at −20˚C. Then, 250 μl lysis buffer (50 mM Tris-HCl, 1% Triton X-100 at pH 8) were added before shaking the plates at room temperature, 1,400 rpm for 60 min. Cell debris was removed via centrifugation at 4˚C, 2,500 rpm for 20 min before transferring 175 μl of the supernatant to a new plate. This plate was centrifuged again at 4˚C, 2,500 rpm for 5 min before 75 μl of the supernatant was added to 10 μl CBMs, which have been washed before as described above. After an incubation at 4˚C, 600 rpm for 60 min, the CMBs were washed twice with 145 μl 50 mM TEA buffer at pH 8.5 on a magnetic plate. Finally, the reaction was started by adding prewarmed 50 μl 50 mM TEA buffer at pH 8.5 with 0.5 mM substrate to the washed CBMs. The reaction plate was sealed with sealing mats and incubated at 37˚C and 600 rpm. After 2 h, 6 h, 18 h, and 48 h, the reaction plate was centrifuged for 2 to 3 min at 2,000 rpm to remove water from the sealing mat, placed on a magnetic plate, and 5 μl of each sample were mixed with 5 μl 1% formic acid. These samples were stored at 4˚C or −20˚C before measuring the products via HILIC-ESI-MS[1].

The HILIC-ESI-MS[1] detection of chitooligosaccharide (COS) and partially acetylated COS (paCOS) products was conducted as described previously [38] with the following adaptations to reduce the time for each run. A shorter ACQUITY UPLC BEH Amide column (1.7 μm, 2.1 × 50 mm; Waters Corporation) was used with a flow rate of 0.8 ml/min, and the column oven temperature was increased to 70˚C. A volume of 2 μl of each sample was injected and separated with the following elution profile: 0 to 0.8 min, 15% to 100% B; 0.8 to 1.1 min, 100% to 15% B; 1.1 to 1.5 min, 15% B. Mass spectra were acquired over a smaller scan range from m/z 650 to 900. The data were first inspected by hand using Data Analysis 4.1 (Bruker) before they were converted to mzML format and analyzed with an in-house python script using pymzML v2.0 [39]. The relative acetate release for the A4 (*rarA4*) was calculated based on the relative amounts of A3D1, A2D2, and A1D3 (*raA3D1*, *raA2D2*, and *raA1D3*, respectively) as follows:

$$rarA4 = raA3D1 + 2 \cdot raA2D2$$

Samples taken after 48 h were additionally measured via MS$_2$. They were separated as described for the MS$_1$ measurement, and time windows were set for the corresponding target masses (A3D1: m/z 789; A2D2: m/z 747), which were fragmented both with an amplitude of 1.2 and an isolation width of m/z 3 using the MRM mode. Fragments were observed in a scan range from m/z 200 to 900. The data were converted to mzML format and analyzed with an in-house python script using pymzML v2.0 [39]. The PA was determined based on the method introduced by Cord-Landwehr and colleagues [40], assuming that mainly b- and y-ions were present.

### Detailed screening of 7 positions on A3D1 substrates

The muteins for 7 positions (N75, W76, Y140, F141, D164, Y168, and H199) were screened by hand using DAAA, ADAA, AADA, and AAAD as substrates as described in the previous section Detailed screening of 7 positions with A4. The HILIC-ESI-MS detection of the paCOS products was performed with a Waters Synapt XS HDMS 4k mass spectrometer coupled to a Waters ACQUITY Premier UPLC System. The oligomers were separated with the same columns and elution profile as described in the previous section Detailed screening of 7 positions with A4 only with the 0.4 min equilibration phase shifted to the beginning of each run. Solvent B and solvent A were used as the wash and purge solvent, respectively. MS$^1$ measurements

were conducted in positive resolution mode in a mass range from m/z 650 to 900 with a scan time of 0.1 s. Mass accuracy was ensured using the proton adduct of leucine enkephalin (m/z 555.2692) as the lock mass in 10-s intervals for 1 s. The capillary voltage of the source was set to 1 kV, source and desolvation temperature to 150˚C and 250˚C, respectively. Cone gas and desolvation gas flow were set to 0 l/h and 750 l/h, respectively, resulting in a nebulizer pressure of 6.5 bar. $MS^2$ spectra were measured using the same conditions in TOF MS/MS mode with target masses set for A3D1 (m/z 789.325), A2D2 (m/z 747.314), and A1D3 (m/z 705.304) with respective collision energies of 22 V, 24 V, and 24 V at the expected elution time. The scan range was set to m/z 100 to 1,000 with an LM resolution of 4.7 for the quadrupole. The sample data were processed as described for A4, and the relative acetate release for the A3D1 substrates (*raA1D3*) was calculated based on the relative amounts of A2D2, A1D3, and D4 (*raA2D2*, *raA1D3*, and *raD4*) as follows:

$$rarA3D1 = raA2D2 + 2 \cdot raA1D3 + 3 \cdot raD4$$

## Polymer *N*-acetylation

Polyglucosamine was *N*-acetylated with PesCDA$^{nm}$, PesCDA H199Y, and PesCDA H199K, which showed an increased and decrease GlcNAc preference at subsite −1 during our SSM library screening, respectively. These were first expressed in 500 ml $LB_{Amp100}$ autoinduction medium [37] for 48 h at 26˚C, 120 rpm. The cells were harvested by centrifugation at 4,000*g* for 20 min at 4˚C, and the pellet was resuspended in 30 ml FPLC buffer (20 mM TEA, 400 mM NaCl at pH 8) and stored at −20˚C. The cell mixtures were thawed at room temperature before adding 3 µl benzonase (Merck KGaA, 25 U/µl) in 250 µl 2 M $MgCl_2$. After 15-min incubation at room temperature, 2 ml high salt buffer (1 M TEA, 1 M NaCl at pH 8) were added before the cells were lyzed by sonication with a Branson Digital Sonifier model 250-D (Emerson) using five 15-s pulses at 40% amplitude. After cell lysis, insoluble debris was pelleted for 60 min at 40,000*g*, 4˚C. The enzymes were purified from the supernatant by affinity chromatography using the Strep-TactinXT purification system (IBA). Then, the enzymes were concentrated with Amicon Ultra-15 centrifugal filters (Merck KGaA) and rebuffered into FPLC buffer. Enzyme concentrations were determined using the Bradford method [41].

After enzyme production, chitosan 134 (1 mg/ml, MW 190 kDa, $F_A$ 0.01) was incubated with 25 µg/ml of each enzyme in 1.5 M sodium acetate buffer at pH 7 at 37˚C. Samples were taken after 0.33 h, 0.66 h, 1 h, 2 h, 3 h, 4 h, 6 h, 8 h, 10 h, 14 h, and 24 h, and the reaction was stopped by polymer precipitation with 1 vol. of acetone and 0.066 vol. of 1% ammonia. All samples were centrifuged for 40 min, 16,000*g* at 4˚C, and the polymer pellet was vacuum dried. The pellet was resuspended in 0.3 vol. water and vacuum dried again 2 times before it was finally resuspended in 1 vol. 200 mM sodium acetate buffer at pH 4.2. The chemical controls were produced with chitosan 134 as the starting material as described previously [42]. The $F_A$ of all samples as well as the chemical controls (15 µg polymer per sample) was determined as described by Wattjes and colleagues [43]. For each chitosan (produced chemically or enzymatically), samples were chosen, which are close to $F_A$ 0.1, 0.2, 0.3, 0.4, and 0.5. These samples were split into 3 reactions to determine the PA via enzymatic mass spectrometric fingerprinting with chitinosanase [44] in triplicates. Samples were measured by HILIC-ESI-MS as described previously [45] and via SEC-RI-ESI-MS. For the latter method, 1.5 µg of the polymer digests were separated at 40˚C with a ACQUITY UPLC Protein BEH SEC column (125 Å, 1.7 µm, 4.6 mm × 300 mm, 1 K to 80 K, Waters Corporation) using 0.4 ml/min of 150 mM ammonium acetate and 200 mM acetic acid in water as the eluent. After separation, the flow was split equally going into an ERC RefractoMax 520 (Thermo Fisher Scientific) and an ESI-MS (amaZon Speed, Bruker). The RI signal was acquired with a rate of 10 Hz, a recorder

range of 512 μRIU and an integrator range of 125 μRIU/V at 40°C, while the ESI-MS was operated as described before [38].

The weight average A- and D-block sizes *block(A)$_w$* and *block(D)$_w$* are calculated as follows:

$$block(A)w = \frac{\sum_i (DP_i \cdot I_i \cdot N(A)_i)}{\sum_i (DP_i \cdot I_i)}$$

$$block(D)w = \frac{\sum_i (DP_i \cdot I_i \cdot N(D)_i)}{\sum_i (DP_i \cdot I_i)}$$

where *DP* is the degree of polymerization, *I* the relative oligomer intensity, *N(A)* the number of GlcNAc units, and *N(D)* the number of GlcN units for each oligomer *i* obtained by chitinosanase digestions.

## Results

For a better understanding of the following results, we use several notations to indicate the substrate, its pattern of acetylation, and, if applicable, we visualize how the substrate was placed in the enzymes binding site. The substrate composition is indicated by the number of A (GlcNAc) and D (GlcN) units. Accordingly, chitotetraose consisting of 4 A units will be denoted as **A4**, while the first deacetylation product will be denoted as **A3D1**. To indicate which of the GlcNAc units was deacetylated, all units will be written out, starting at the nonreducing and proceeding to the reducing end, such as **AADA**. In this example, the A4 substrate had been bound to the enzyme from subsite −2 to subsite +1 (with subsite 0 being the one where deacetylation occurs), which will be written as **binding mode [−2,+1]**. If we refer to a specific substrate in a certain binding mode, this will be indicated with a lower case letter for the sugar unit bound at subsite 0. As an example, A4 in binding mode [−2,+1] will be written as **AAaA**.

It should be noted, that the term "subsite" does not necessarily imply a significant interaction between the corresponding region in the binding site and the closest sugar unit. Instead, it refers to the region closest to the indicated sugar unit relative to subsite 0 regardless of whether or not any significant interactions occur.

### SSM library preparation and generation

PesCDA was previously shown to deacetylate A4 at the third position from the nonreducing end in binding mode [−2,+1], producing AADA [4]. Thus, to identify interesting residues, potentially interacting with the substrate, a chitin tetramer (AAaA) was docked in silico into a PesCDA homology model in binding mode [−2,+1] (see Fig 1). Based on their close proximity to the substrate, 27 residues were chosen for mutational studies. These are either part of the conserved motifs (MTs) or 6 loops (L1 to L6) surrounding the binding site defined by the subsite capping model [30]. Both catalytic residues D46 from MT1 and H196 from MT5 were included as negative controls. Therefore, the metal binding residues from MT1 and MT2 were not included as their muteins were expected to be mainly inactive as well. From MT2, only S102 was chosen due to its close proximity to the substrate. Further residues are from L1 (Q74 to S78), MT3 and part of L3 (R137 to S142), MT4 and part of L4 (I163 to H170) as well as MT5 and the beginning of L6 (L194 to H199). For some of these residues, such as the catalytic aspartate (D46) and histidine (H196) and their supporting residues arginine (R137) and aspartate (D167), their function has been well described for other CDAs such as ClCDA [34]. These also include S102 from MT2, the hydrophobic pocked forming leucine from MT5 (L194) and the main chain of Y140 from MT3. However, for the latter, the role of the residue's side chain was

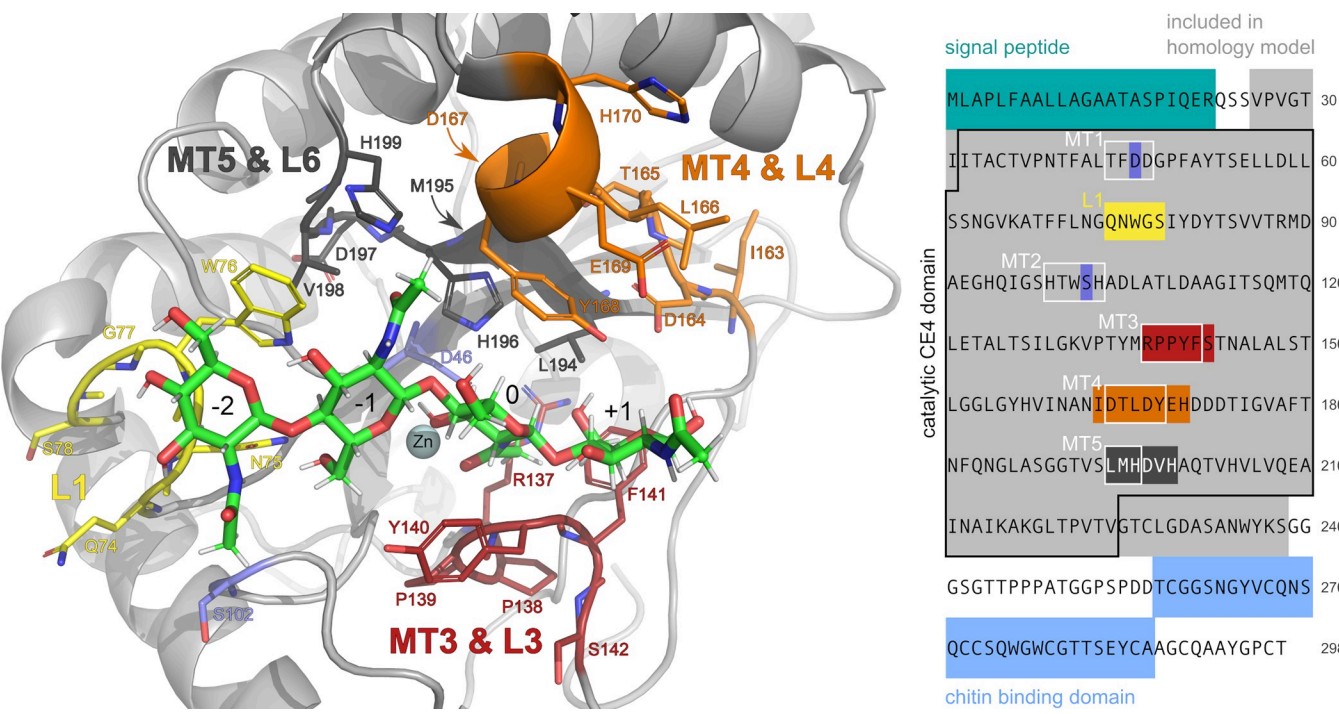

**Fig 1. Amino acids in close proximity to AAaA in binding mode [−2,+1], which were chosen for the SSM library.** In the structure, the substrate is shown with green carbon atoms and the sugar units are labeled according to their subsite. Loops (L1, L3, L4, and L6 according to Andres and colleagues [30]) and motifs (MT3, MT4, and MT5) are color-coded and the corresponding residues are labeled accordingly. Hidden residues are indicated by a small arrow. Single residues from MT2 (D46) and MT2 (S102) are highlighted separately in blue. In the amino acid sequence on the right (UniProt ID: A0A1L3THR9), the chosen residues are colored accordingly. In addition, the 5 CDA motifs (MT1–MT5), the catalytic CE4 domain, the signal peptide, and the chitin binding domain are indicated.

only briefly described for SpPgdA [22], even though it is highly conserved in many CDAs. To our knowledge, the structural or functional role of the remaining residues has not been investigated in detail yet, even though some seem to be conserved in many CDAs.

The plasmid used for library generation was based on an available construct [4], replacing the CBDs of PesCDA by the CBD 1 and 2 of VcCDA, as the screening method used was originally designed for VcCDA [27]. We aimed to mutate all 27 chosen positions to all non-wild-type amino acids, resulting in a library size of 513 muteins of which 470 were successfully verified by sequencing.

## First screening of all 27 positions

This SSM library was screened according to Pascual and Planas [27] with some modifications (see sec. Complete library screening with A4 for more details). In short, the *E. coli* cells expressing the different muteins or the nonmutated construct termed "PesCDA[nm]" were grown, harvested, and lyzed in 96-well plates. The target enzymes were purified using small amounts of chitin-coated magnetic beads, based on preparatory test (see S1 Fig), aiming at saturating the beads to ensure similar enzyme concentrations in all wells. Then, the substrate was added and the plates were incubated at 37°C for 2 h before the amount of free primary amines was detected using a fluorescamine-based method. The muteins' activities were normalized to PesCDA[nm] present in each plate (Figs 2 and S3).

It should be pointed out that we evaluated the results carefully, avoiding to rely on the activity levels of a single mutein without further rationalization. Due to the large number of

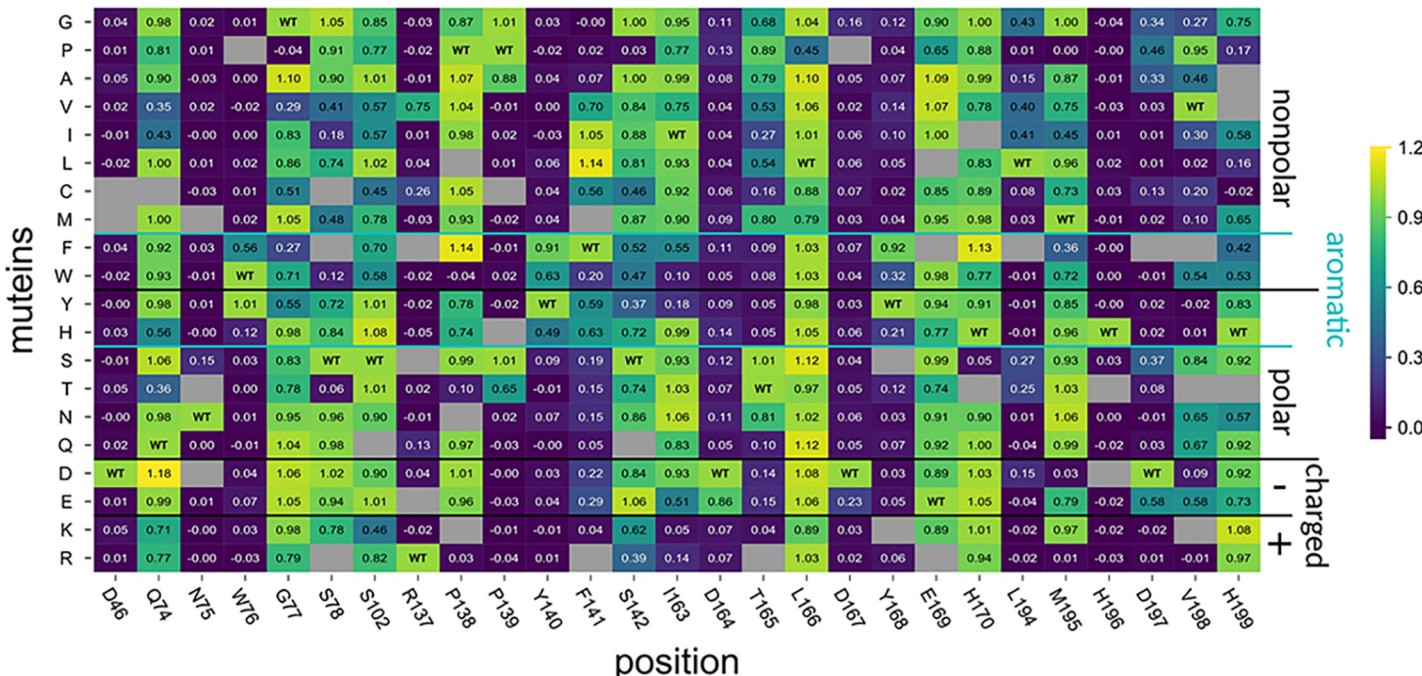

**Fig 2. Activity of PesCDA muteins on A4 normalized to PesCDA^nm estimated by fluorescence-based quantification of free primary amines.** Each column shows the activity of all available muteins at a given position, as indicated below the matrix. Missing muteins are grayed out. For orientation, PesCDA^nm is included in each column at the corresponding position of the wild-type (WT) amino acid. The muteins are grouped by the properties—nonpolar, polar, charged (+/−), or aromatic—of the residue by which the WT amino acid was exchanged. Corresponding standard deviations can be found in a separate heatmap in S3 Fig (*n* = 3–4). All values can be found in S1 Data.

muteins and because the library generation was mainly done manually, the results are likely to include some false positive or false negative results. Furthermore, despite the small amount of chitin-coated beads used to ensure saturation and, therefore, equal amounts of enzymes in all wells, it is still possible that some muteins were misfolded or expressed with very low yields. Consequently, the screening is not capable of distinguishing between these cases; however, if a mutation likely results in misfolding and, thus, a reduced activity, this will be further elaborated below. First, we looked at the overall activity pattern for each position. Based on these patterns and to allow better understanding of the results, the positions tested are sorted into the following 4 groups:

**Group 1 (D46, N75, R137, D167, H196):**
All muteins show no or strongly reduced activity.

**Group 2 (W76, Y140, Y168):**
Only muteins with aromatic residues show good or even PesCDA^nm activities.

**Group 3 (P139, F141, I163, D164, T165, L194, D197, V198, H199):**
A group of active or inactive muteins share a common characteristic of the muteins' residues.

**Group 4 (Q74, G77, S78, S102, P138, S142, L166, E169, H170, M195):**
Muteins' residues of muteins with reduced or no activity do not share a common characteristic and most muteins show PesCDA^nm activity levels.

As expected, both catalytic residues (D46 and H196) can be found in group 1 with all muteins being inactive. This also applies to R137 stabilizing the catalytic D46, and D167, which increases the pK_a value of the catalytic histidine. The seemingly high activity level of

R137V is possibly a false positive result, as valine is not expected to compensate for the missing arginine. Group 1 also contains N75 as part of an elongated L1, which can be found in several fungal CDAs, all containing an asparagine in this position [5,34,46,47]. Another residue of L1, namely W76, is present in group 2, where only aromatic residues can substitute the wild-type (WT) amino acid. A similar pattern can be found for Y140 and Y168, suggesting that the aromatic character of these group 2 residues plays an important role for substrate binding. The activity patterns for both group 1 and group 2 suggest that these residues play an important role for enzyme activity, either by participating in the catalytic mechanism or by substrate binding. A more detailed analysis of their role will be elaborated in the next section In silico studies of PesCDA.

Residues that belong to group 3 can be replaced with several other residues, still resulting in active enzymes; however, some or even most substitutions strongly reduce their activity. This applies to P139, which is conserved among most CDAs and can only be replaced by very small amino acids, indicating spatial constraints at this position. F141 can be replaced by larger hydrophobic residues and, to a certain extent, by other aromatic residues. This could match the role of the aligning leucine in ClCDA, which forms a hydrophobic pocket for the acetyl methyl group at subsite 0 [34]. The opposite site of this hydrophobic pocket is formed by L194, which cannot be replaced without losing activity in this screening. The 3 consecutive positions I163, D164, and T165 as part of MT4 also show distinct activity patterns. D164 can only be substituted with the other negatively charged amino acid without strongly losing activity, indicating that the negative charge is needed for enzyme activity, likely for enzyme substrate interaction as further discussed in the following section. In contrast to D164, the side chains of the 2 neighboring residues I163 and T165 point towards the inside of the enzyme (Fig 1), probably resulting in misfolding if the substituted amino acids are too big or charged. The last 3 residues from group 3 are part of L6 following MT5. D197 cannot be replaced by any amino acid without losing activity and most muteins are even inactive. It seems that only a glutamate can somewhat replace the WT residue, while the other substitutions that still result in activity are very small amino acids, likely to not interfere with protein folding at this position. The activity pattern at position V198 seems surprising as some muteins show activity levels above 0.5 (i.e., half of the PesCDA$^{nm}$'s activity), whereas this is not the case for similar residues such as isoleucine or leucine. The high activity of V198P hints at a structural function where the proline is the only other amino acid that facilitates the tight turn at this position, whereas other amino acids likely disturb the folding to different extents. H199 is another interesting position with only a few nonpolar amino acid substitutions strongly reducing the activity at this position, while especially polar and charged residues retain PesCDA$^{nm}$ activity levels. As this position shows additional interesting activity patterns in further analyses, the role of this position will be discussed in more detail below.

All positions that belong to group 4 show PesCDA$^{nm}$ activity levels for most of their muteins. The 3 positions Q74, G77, and S78 form the beginning and end of L1 and are all surface exposed residues, seemingly not crucial for the correct folding of this elongated loop. The same applies to P138 in MT3, S142 in L3 as well as E169 and H170 at the beginning of L4. Even though the remaining 3 residues S102, L166, and M195 are all enclosed by highly conserved residues of MT2, MT4, and MT5, respectively, they seem to have a negligible impact on protein activity. The activity patterns observed for all positions in this group indicate that the WT amino acids do not have a distinct feature necessary for enzyme activity. Only a few substitutions reduce enzyme activity, suggesting a negative effect of these substitutions rather than a missing positive effect caused by the WT amino acids.

Using this screening method, which gave valuable insight into the muteins' activities on the fully acetylated chitin tetramer A4, we screened a subset of the SSM library using the mono-

deacetylated chitosan tetramer ADAA as well. However, due to the high background fluorescence of the substrate and the comparatively low fluorescence increase upon enzyme activity, the results showed large standard deviations making them very difficult to evaluate. Furthermore, PesCDA$^{nm}$ has a reduced activity on ADAA compared to A4, and, thus, testing different reaction times would be needed to find activity patterns that could be compared properly between the 2 substrates. Thus, we decided to stop the reaction at different times and analyze the products via high pressure liquid chromatography–mass spectrometry (HPLC-MS). As this approach requires dramatically increased analysis time, we first evaluated the experimental results obtained so far in more detail using in silico tools, to further narrow down the most interesting positions to be eventually screened with A4 as well as several mono-deacetylated A3D1 substrates.

### In silico studies of PesCDA

Although PesCDA mainly produces AADA [4], upon longer incubation times, AADA is bound in binding mode [−1,+2] producing ADDA with a low conversion yield [48]. In the presence of excess acetate, D4 is *N*-acetylated to DDAD (binding mode [−2,+1]); however, this is then further converted to DDAA as a result of binding mode [−3,0] [48]. Based on these 3 different binding modes observed in vitro, A4 was docked into a PesCDA homology model in binding mode [−2,+1] (AAaA), [−1,+2] (AaAA), and [−3,0] (AAAa). To probe the effect of GlcN units at all potential subsites, all A3D1 substrates (DAAA, ADAA, AADA, and AAAD) were docked in the same binding modes as well. Even though not all of them are expected to bind in all 3 binding modes, especially for the latter 2 binding modes, all of them were included here. For each substrate and binding mode, 3 conformations with the highest docking scores were chosen and subjected to MD simulations.

First, we calculated the RMSF as well as the binding free energy (BFE) for each sugar unit at the corresponding subsite and the average RMSF and total BFE for the whole tetramer (Fig 3). The BFE was calculated using the MMGBSA approach. In this method, the van der Waals, electrostatic, polar, and nonpolar solvation energies are calculated for the receptor and ligand separately and for the whole complex. The resulting delta between receptor plus ligand and the complex is an estimate for the substrates' BFE. While the RMSF nicely illustrates the movement at each subsite, the BFE calculation allows a more detailed understanding of the substrate binding. We would like to point out that these calculations are not meant to yield realistic kcal/mol numbers but instead yield BFE values that can be compared between different simulations using a similar setup. For those readers interested in the comparison between in silico and in vitro results, we have a detailed discussion about this in our recent paper about the *Aspergillus niger* CDA (AngCDA) [32]. In the following, we will highlight some of the observations from our MD simulations and relate them to the calculated RMSF and BFE values.

For A4 in binding modes [−2,+1] and [−3,0], both the average RMSF and the total BFE are lower compared to binding mode [−1,+2]. The presence of a GlcN unit at subsite −3, −2, and 0 seem to increase both values for all binding modes while for a GlcN units at subsite −1 and +1, no clear tendency is visible.

Taking a look at the local RMSF and BFE for A4 in all binding modes, the sugar units at subsite 0 show the lowest fluctuation and binding energy while it tends to increase further away from subsite 0 with the strongest movement and weakest binding at subsite +2. This indicates stronger/more interactions between the substrate and the enzyme at the center of the binding groove and fewer interactions especially at subsite +2. Again, the presence of GlcN units influences both the RMSF and BFE. As expected, a GlcNAc unit at subsite 0 contributes most to substrate binding. Here, the acetyl group points towards the binding groove enabling

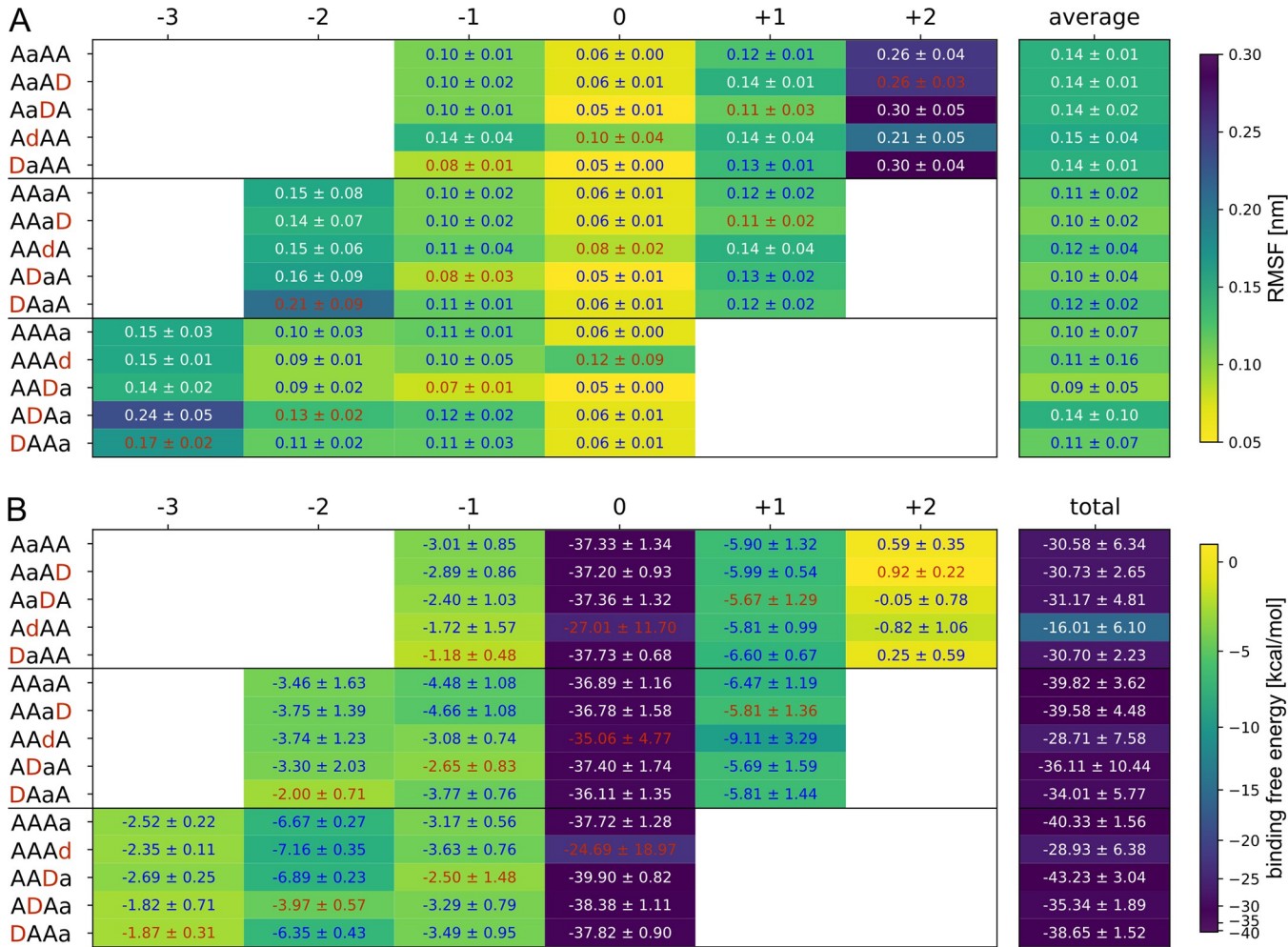

**Fig 3. Substrate fluctuation and binding energy calculated from MD simulations.** (**A**) Root mean square fluctuation (RMSF) of the different substrates and their indicated individual sugar units in the given binding modes. (**B**) Binding free energy (BFE) contribution of the different substrates and their indicated individual sugar units in the given binding modes. The values for GlcN units are highlighted in red. All values are means ± SD of 9 replicates based on 3 different starting structures each. All values can be found in S2 Data.

strong interactions with the enzyme. As the acetyl group is missing in GlcN units, they show a higher fluctuation and BFE at subsite 0, which also influences the neighboring subsites (see AdAA and AAdA). Similar to subsite 0, the acetyl group at subsite −2, if present, points towards the binding site. Although a GlcNAc unit at the nonreducing end always shows a higher fluctuation and higher BFE compared to the neighboring unit, its BFE is lowest if bound to subsite −2. Furthermore, the GlcNAc unit at subsite −2 in binding mode [−3,0] shows the lowest BFE in all nonzero subsites, highlighting the importance of this subsite for substrate binding if a GlcNAc unit is placed here. Contrary to subsite −2 and 0, the acetyl group at subsite −1 and +1 points away from the binding groove, potentially allowing stronger fluctuations of GlcNAc units. Indeed, at both subsite −1 and +1, the RMSF of GlcNAc units is increased compared to GlcN units. However, the BFE of GlcNAc units is increased at subsite −1, whereas it is similar to GlcN units at subsite +1. This suggests that the acetyl group at subsite −1 still seems to interact with the enzyme to an extant that lowers the BFE of GlcNAc units, whereas such an interaction is missing at subsite +1. Consequently, our in silico data so

far hint at a subsite preference for GlcNAc units at subsite −2, −1, and, potentially, −3, while subsite +1 seems to have no preference.

The total BFE calculates all interactions between the enzyme and its substrate, whereas the local BFE values solely reflect the contribution of the different sugar units, not representing the contribution of the amino acids they interact with. Thus, we also carried out a decomposition analysis, giving the BFE for each residue to take a closer look at their contribution to substrate binding.

According to the MMGBSA decomposition analysis (Fig 4A and S2 Data), the catalytic D46 and all 3 metal binding residues D47, H99, and H103 contribute strongly to substrate binding in our MD simulations, mostly independent of the binding mode. However, the BFE differs if a GlcN unit is placed at subsite 0, especially for D47, which has the lowest BFE. To get a better understanding of substrate–residue interaction, we further evaluated the most prominent hydrogen bonds at each subsite (Fig 4B and S2 Data). Here, the oxygen of D47, not

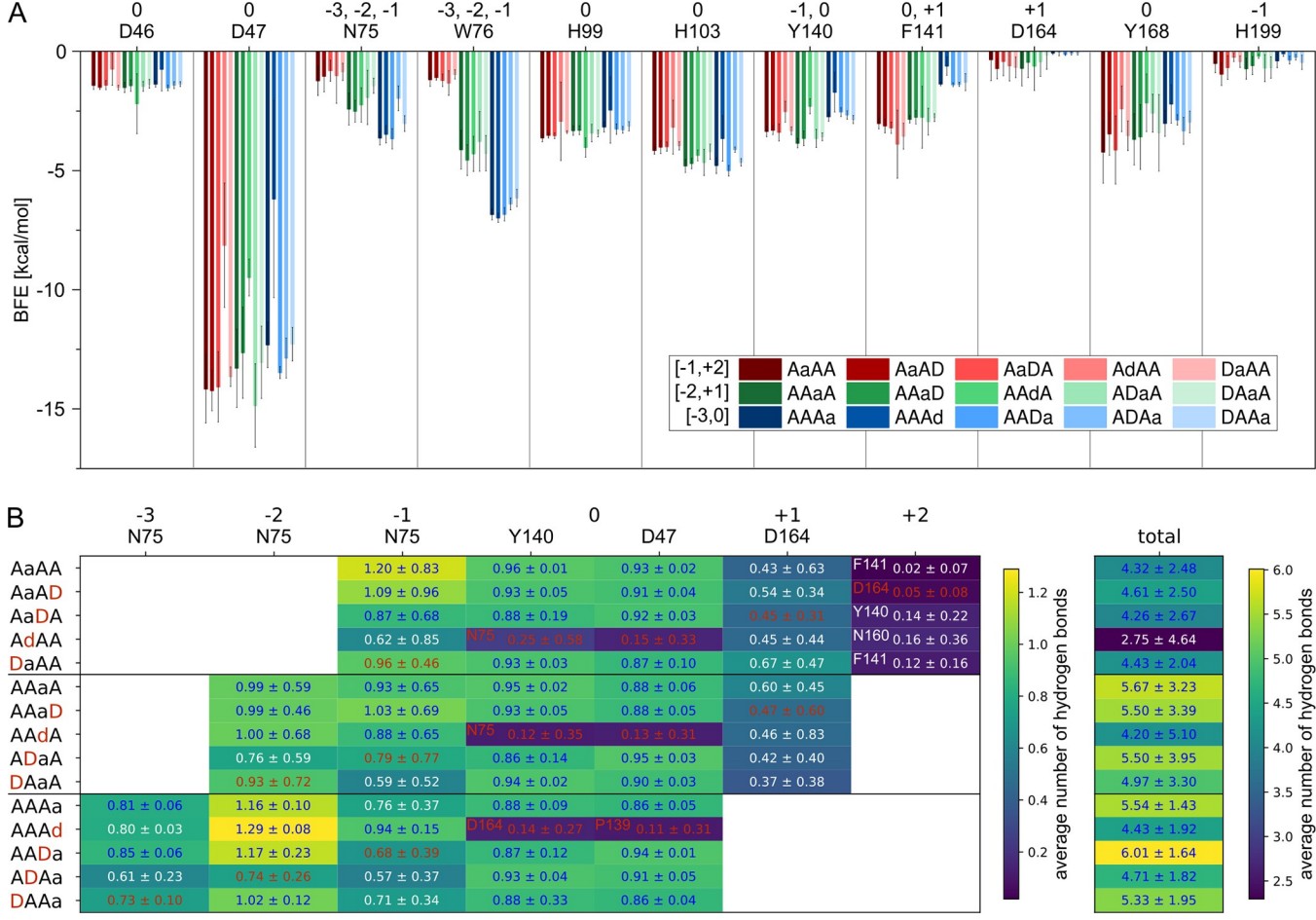

**Fig 4. Binding energy contribution and number of hydrogen bonds of binding site residues.** (**A**) Average binding free energy (BFE) of key amino acids contributing to substrate binding. The amino acids and the subsites, to which they can be accounted, are shown on top. The bars are color coded according to the binding mode and substrate as indicated in the figure legend, where the sugar unit, which is placed at subsite 0, is highlighted with a lower case letter. (**B**) Average number of hydrogen bonds between the amino acid and the sugar unit at the indicated subsite (left) and the total average number of hydrogen bonds between PesCDA and the indicated substrate (right). The amino acid shown on top has the highest number of hydrogen bonds at the indicated subsite unless another amino acid is shown in the corresponding box. The values for GlcN units are highlighted in red. All values are means ± SD of 9 replicates based on 3 different starting structures each. The BFE and hydrogen bond occupancy values for all binding site residues, including all SSM library residues, can be found in S2 Data.

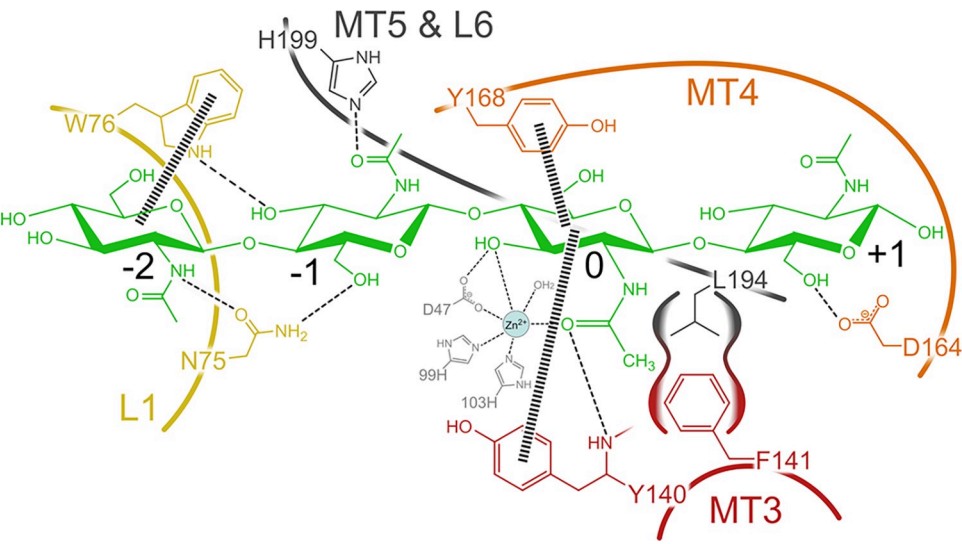

**Fig 5. Schematic drawing of PesCDA substrate interactions with AAaA.** Hydrogen bonds and interactions with the metal ion are indicated with slim dashed lines, while carbohydrate/aromatic interactions are indicated by wide dashed lines. Residues that are located in the same loop (LX) or motif (MTX) are shown in the same color and connected by a curved line. The hydrophobic pocket formed by F141 and L194 is indicated by an extra outline. The color scheme is adapted from Fig 1.

coordinating the metal ion, forms a very stable hydrogen bond at subsite 0 in case it interacts with a GlcNAc unit (Fig 5). The same applies to Y140 of which the main chain nitrogen forms a stable hydrogen bond with the acetyl oxygen, if present, at subsite 0. Although it is noteworthy that the electrostatic interaction is the strongest contributor for the main chain, it is completely compensated by the positive polar solvation energy. Thus, the side chain van der Waals interactions, which are not compensated by a positive polar solvation energy, mostly account for the strong BFE of Y140. An even higher polar solvation energy causes the surprisingly high BFE values for N75, despite that fact that it forms the most stable hydrogen bonds at subsites −3, −2, and −1. The side chain nitrogen and oxygen form hydrogen bonds with the −1 sugar in binding mode [−1,+2], whereas in binding mode [−2,+1], the oxygen and nitrogen mainly interact with the −1 and −2 sugar units, respectively (see Fig 5). In binding mode [−3,0], the N75 main chain oxygen hydrogen bonds with the −3 sugar O6. This increasing number of hydrogen bonds results in a decreasing BFE of this residue from binding mode [−1, +2] to [−2,+1] and [−3,0]. A comparison of AAaA with DAaA as well as AAAa with ADAa shows that a GlcN unit placed at subsite −2 does not only reduce the number of hydrogen bonds with N75 at subsite −2 but also at the neighboring ones. This correlates with the increased RMSF for these substrate bindings (Fig 3A) indicating that N75-GlcNAc interaction at subsite −2 strongly stabilizes the binding at the minus subsites. The second prominent residue interacting at the minus subsites is W76, which contributes most via carbohydrate/aromatic interactions with the −2 sugar unit. As for N75, the BFE decreases if more minus subsites are occupied. Besides W76, both F141 and Y168 show mainly hydrophobic interactions with the substrate as well. Similar to Y140, Y168 stacks with the subsite 0 sugar unit, although its hydroxy group points towards the plus subsites and not the minus subsites (see Figs 1 and 5). As for Y140, its BFE increases if a GlcN unit is present at subsite 0, even though this cannot be explained by a missing hydrogen bond but rather the increased fluctuation of GlcN units at subsite 0. Different to the other aromatic amino acids, F141 does not interact with the A4 substrate by stacking against the sugar but it forms a hydrophobic pocket together

with an opposing leucine for the acetyl methyl group at subsite 0. Thus, the BFE in binding mode [−3,0] increases as F141 is more solvent exposed. Surprisingly, its contribution is not decreasing if a GlcN is bound at subsite 0 in binding modes [−1,+2] and [−2,+1]. In these binding modes, it appears to stack against the +1 sugar unit made possible by a higher flexibility of the subsite 0 GlcN unit, which seems to correlate with the low BFE at subsite +1 in binding mode [−2,+1] (Fig 3B).

The results discussed so far suggest to screen N75, W76, Y140, F141, and Y168 in more detail as they seem to influence the substrate binding most strongly, while not being directly involved in metal ion coordination or catalysis. However, only N75 and W76 do not interact mainly with the subsite 0 sugar unit, possibly preferring a GlcNAc or GlcN unit. To include more residues that could influence subsite preferences, we took a further look at additional residues at all nonzero subsites. According to the hydrogen bond analysis, both oxygens from D164 at subsite +1 form hydrogen bonds with the +1 sugar O6 during approximately half of the MD simulations depending on binding mode and substrate (Figs 4B and 5). Despite the high polar solvation energy for D164 resulting in only a slightly negative BFE, we assume that this residue is important for substrate binding at subsite +1, partly resulting in the low BFE of the +1 sugar unit (Fig 3B). Because the hydrogen bond is not formed with the acetyl group, the binding energy at subsite +1 seems to be mainly independent from whether a GlcNAc or GlcN unit is present. At subsite −1, N75 forms the most stable hydrogen bond during MD simulations but, similar to D164, only with the O6 and occasionally with the O3 atoms of the sugar. At this subsite, H199 hydrogen bonds with the acetyl oxygen of the −1 GlcNAc unit (see Fig 5). According to the MD simulations, this hydrogen bond is solely present in 1 out of 4 frames. Although because the protonation states of histidines are difficult to predict and highly pH dependent at physiological pH, this hydrogen bond might be more or less stable in in vitro experiments.

We did not expect that the residues discussed here all influence subsite preferences or could be mutated to do so. Still, we chose to screen the SSM library positions N75, W76, Y140, F141, D164, Y168, and H199 to learn more about their muteins and, thus, their role in substrate binding and possibly subsite preferences in PesCDA.

## HPLC-MS-based screening of 7 chosen positions

### Mutein screening with A4

The SSM library, with all muteins from the 7 positions chosen based on the in silico analysis, was screened using A4 as a substrate. As described in section First screening of all 27 positions, the muteins and PesCDA[nm] were expressed and purified in 96-well plates before incubating them with A4 at 37°C. Here, however, reaction samples were taken after 2 h, 6 h, 18 h, and 48 h to get insights into the product development over time. Unlike the first screening, where free primary amines were detected photometrically, the products were detected by HPLC-MS allowing the distinction between the first and second deacetylation product A3D1 and A2D2, respectively. For a comparable representation to the first screening results (Fig 2), we calculated the amount of free primary amines based on the MS results for all 4 incubation times and plotted the muteins' activities normalized to PesCDA[nm] (Figs 6 and S4). We would like to highlight again that we tried to focus not solely on single activity values but compared several muteins with each other and looking for certain trends that correlate with the muteins amino acid properties. If single muteins are highlighted in more detail, we tried to connect the observations from our screening with the MD simulations.

Although some muteins show higher or lower activities, the general trends for the 2-h results compared to the initial screening results (Fig 2) remain the same. The following time

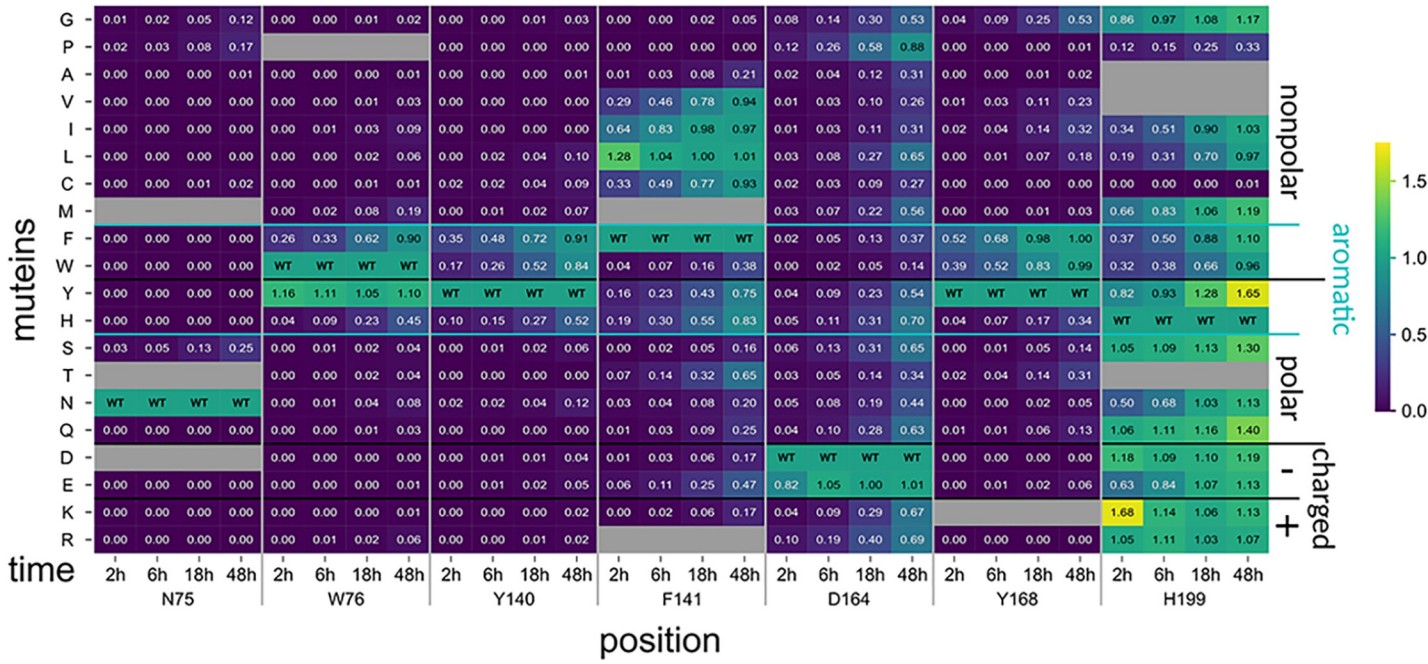

**Fig 6. Subset of PesCDA muteins activity on A4 normalized to PesCDA$^{nm}$ estimated by product detection via HPLC-MS.** Each column shows the activity of all available muteins at one position for the indicated incubation time. Missing muteins are grayed out. For orientation, PesCDA$^{nm}$ is included in each column at the corresponding position of the wild-type (WT) amino acid. The muteins are grouped by their properties—nonpolar, polar, charged (+/−), and aromatic—of the residue by which the WT amino acid was exchanged. Corresponding standard deviations can be found in a separate heatmap in S4 Fig (*n* = 4). All values can be found in S3 Data.

points reveal some differences that were not visible in the initial screening. First, they allow to further distinguish muteins with low to no activities. All muteins of N75 except for N75S seemed to be completely inactive; however, over time, N75G and N75P also deacetylated the substrate to a small extent, whereas the remaining muteins appear indeed to be inactive. This is also the case for the group 2 positions W76, Y140, and Y168, for which the nonaromatic substitutions seem to strongly reduce or mostly even eliminate enzyme activity. However, the data reveal that some muteins were able to deacetylate up to 50% of the A4 substrate after 48 h, such as aliphatic and very small amino acid substitutions for Y168. This also suggests that Y168 might be less vital for enzyme activity compared to N75, W76, and Y140, where substitutions that are not similar to the WT residues almost or even completely inactive PesCDA. This effect is even stronger for the D164 position, where all muteins increasingly show activity over time, starting to converge towards PesCDA$^{nm}$ after 48 h. This effect of drawing level with PesCDA$^{nm}$ can be explained by its product production over time shown in Fig 7A (see S3 Data for all values). PesCDA$^{nm}$ quickly converts A4 to A3D1 almost completely within 6 h, equivalent to a relative acetate release of 0.89, whereas it produces only low amounts of A2D2 even after 48 h. During this time, some of the less active muteins such as W76F, Y140F, and Y168F are able to converge towards PesCDA$^{nm}$. This effect also applies to aliphatic substitutions for F141 (F141V/I in Fig 6), except for F141L, which seems to be even more active than PesCDA$^{nm}$ after 2 h. Still, it does not exceed final A3D1 and A2D2 levels produced by PesCDA$^{nm}$. This is different for H199K, which is even more active after 2 h and produces measurable amounts of A2D2 already after 6 h, exceeding relative acetate release levels of 1. However, as for PesCDA$^{nm}$, the curve flattens out, indicating that more A2D2 is solely produced during the experiment because of the strong activity increase. Other H199 muteins such as H199Y,

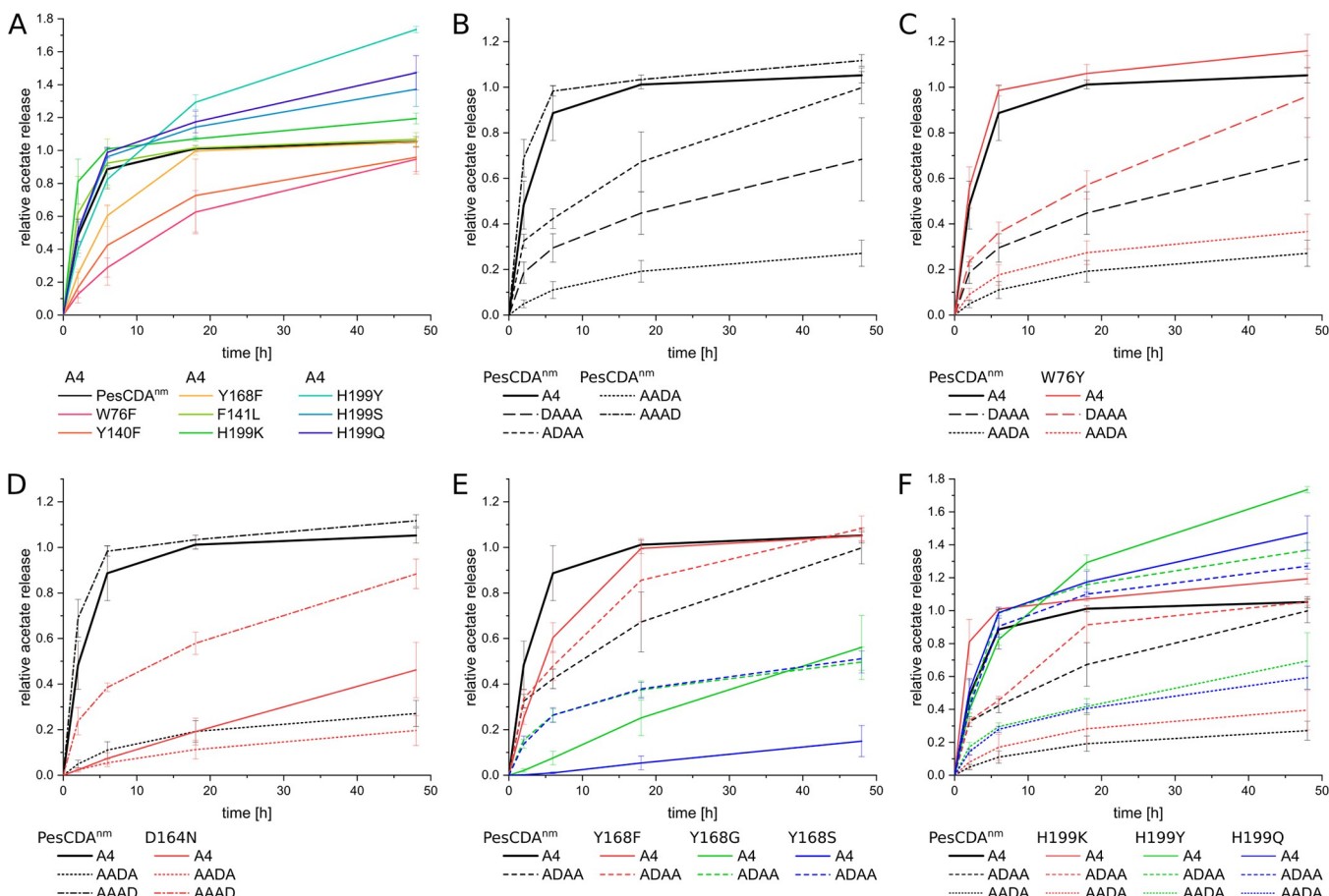

**Fig 7. Relative acetate release of PesCDA$^{nm}$ and selected muteins on A4 and A3D1 substrates.** The curves show the acetate released from A4 relative to the substrate concentration. (**A**) Comparison of PesCDA$^{nm}$ and different muteins' activities on A4. (**B**) Comparison of PesCDA$^{nm}$ activities on A4 and all A3D1 substrates. (**C, D**) Comparison of PesCDA$^{nm}$ and selected muteins of one position on A4 and A3D1 substrates (WT: $n = 28$; muteins: $n = 4$). All values can be found in S3 Data.

H199S, and H199Q, however, show a deviating curve progression, producing even more A2D2 than the highly active mutein H199K by the end of the experiment, despite their activities similar to PesCDA$^{nm}$ after 2 h. Whereas PesCDA$^{nm}$ started to produce low amounts of A2D2 only after A4 is almost completely converted to A3D1, H199Y already produced A2D2 when significant amounts of A4 were still present.

In addition to the MS$^1$ measurements presented in this section, the 48-h samples of all muteins were analyzed by MS$^2$ as well, fragmenting the A3D1 and A2D2 products (see S4 Data). Based on the fragment composition, the PA of these products can be deduced as described previously [40]. It should be noted, however, that the products were not chemically *N*-acetylated nor labeled at the reducing end, making the pattern determination less robust. Still, differences to the PesCDA$^{nm}$ are clearly visible if present. As expected, the main pattern for A3D1 is AADA with 95% for PesCDA$^{nm}$ and 83% to 98% for the muteins, if more than 5% A3D1 were produced. Of those muteins producing less than 90% AADA, muteins of position Y168 produce more ADAA, while muteins of positions F141 and D164 produce increased amounts of AAAD. Both indicate a small change in binding mode, which is especially striking for F141K and F141Q with the lowest amount of AADA and the highest amount of AAAD. This effect will be further discussed in the next section.

When deacetylating A4, the A2D2 pattern could only be determined for those muteins which produced A2D2 in sufficient amounts. These muteins were further tested on AADA directly, as described in the next section, and revealed very similar A2D2 patterns both on A4 and AADA. Thus, the corresponding $MS^2$ results are presented in the following section, in which the activity of selected muteins was tested on A3D1 substrates.

### Mutein screening with A3D1

As done for the muteins from the 7 positions screened on A4 using the MS-based detection method, the chosen muteins from these positions were additionally screened for their activity on DAAA, ADAA, AADA, and AAAD. We assumed that the deacetylated sugar unit does not strongly influence the binding mode, consequently producing DADA, ADDA, and AADD on the 3 substrates DAAA, ADAA, and AAAD, respectively. Thus, for the corresponding substrates, muteins of those positions were chosen, which are in close proximity to the GlcN unit during the first deacetylation. This differs for AADA, which has to bind in a binding mode other than [−2,+1] for deacetylation. Accordingly, selected muteins from all subsites were tested for their activity on AADA.

Before taking a closer look at the muteins' activities, we first compared the curve progressions for PesCDA$^{nm}$ on all substrates (Fig 7B). As mentioned above, the acetate release on A4 increases quickly in the beginning, starting to plateau after 6 h. A similar progression with an even steeper ascent could be observed on AAAD. The other A3D1 substrates show a slower increase in the beginning, which continues until the end of the experiment, flattening out more slowly compared to A4 and AAAD. Besides showing that a GlcN unit certainly influences the activity dependent on the subsites at which it is placed, these different curve progressions should be kept in mind for further evaluation of the muteins' activities. As discussed in section Mutein screening with A4, due to the plateau reached during the second half of the experiment, less active muteins are able to converge towards PesCDA$^{nm}$ when deacetylating A4. This does not apply to the substrates DAAA, ADAA, and AADA, for which PesCDA$^{nm}$ did not reach a plateau yet. Thus, muteins are not expected to converge towards this plateau in a similarly strong way. $MS^1$ data for the 48-h samples confirm that AAAD and ADAA are mainly converted to AADD and ADDA, respectively, placing the GlcN unit at the expected subsite. The A2D2 product of DAAA could not be safely identified as DADA by our automated script, but we assume that this results from the missing label at the reducing end and that, indeed, DADA is produced. If AADA gets deacetylated, it cannot productively be placed in binding mode [−2,+1]. Here, our $MS^2$ data show that PesCDA$^{nm}$ produces about one-third AADD and two-thirds ADDA, preferring to place a GlcNAc unit at subsite −1, even though this leaves the important −2 subsite unoccupied.

Muteins of position N75 were tested for their activity on DAAA and ADAA, placing the GlcN unit at subsite −2 and −1, respectively (S6 and S7 Figs). Very similar to the activity on A4, most muteins are inactive on these A3D1 substrates, and only N75G, N75P, and N75S show very low activities. Apparently, none of these muteins are able to increase the activity if a GlcN unit is present.

Muteins of the neighboring position W76 were screened on DAAA and AADA (Figs 8 and S5). Most normalized mutein activities were similar for all substrates tested, and some small differences can be attributed to the different curve progressions on A4 compared to DAAA and AADA (see W76F and W76H). Solely, W76Y stands out, which seems to be more active on DAAA and AADA compared to A4. However, it also showed a slightly stronger acetate release on A4, though less visible since both W76Y and PesCDA$^{nm}$ quickly reach a plateau. Because the activity is lower on DAAA and AADA, this plateau is not reached, making these

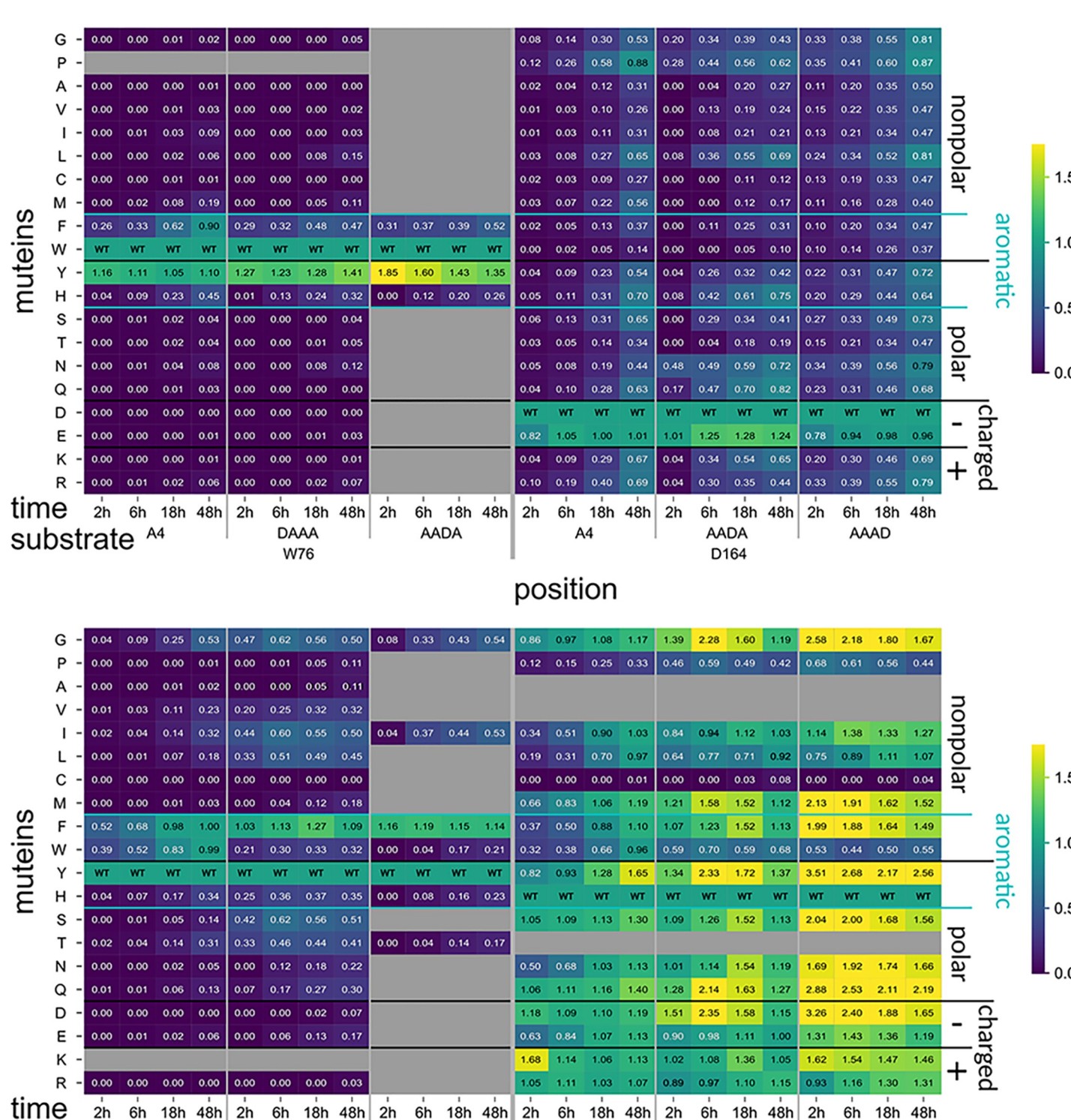

**Fig 8. Subset of PesCDA muteins' activity on A4 and different A3D1 substrates normalized to PesCDA$^{nm}$ estimated by product detection via HPLC-MS.** Each column shows the activity of all tested muteins at one position for the indicated incubation time and substrate. Missing or not tested muteins are grayed out. For orientation, PesCDA$^{nm}$ is included in each column at the corresponding position of the wild-type (WT) amino acid. The muteins are grouped by their properties—nonpolar, polar, charged (+/−), and aromatic—of the residue by which the WT amino acid was exchanged. Corresponding standard deviations can be found in a separate heatmap in S5 Fig (*n* = 4). All values can be found in S3 Data.

small differences more visible (Fig 7C). Thus, we assume that W76Y is slightly more active compared to PesCDA$^{nm}$ on all substrates without influencing subsite preferences.

The second position of group 2, Y140, as well as the neighboring F141 show no difference in their muteins' normalized activity on A4 and AADA (S6 and S7 Figs). It appears that the active muteins reach higher activity levels compared to PesCDA$^{nm}$ when deacetylating A4, although this may be attributed to the muteins converging towards PesCDA$^{nm}$ as mentioned before. These results indicate that neither Y140 nor F141 significantly influence the preference for a GlcNAc or GlcN unit. However, mutating F141 seems to influence the binding mode. While F141V, F141I, F141L, and F141Y produce similar or lower amounts of AADD ($\geq$33%), muteins such as F141Q, F141K, F141E, and F141S produce more than twice the amount of AADD when deacetylating AADA compared to PesCDA$^{nm}$. This strong shift towards binding mode [−3,0], leaving subsite +1 unoccupied seems to occur if the hydrophobic pocket formed by F141 and L194 is disrupted. This could be explained by folding deviations due to the missing hydrophobic interaction between F141 and L194. However, our in silico data suggest that solvent exposure of the hydrophobic pocket reduces the contribution of F141 (see binding mode [−3, 0] in Fig 4). Consequently, if F141 is replaced by a polar or charged residue, solvent exposure of this residue would have a weaker effect on the binding mode.

According to our MD simulations, D164 forms a hydrogen bond with the sugar O6 at subsite +1. Consequently, no direct effect on GlcNAc or GlcN preference was expected. However, it seems that most muteins show an increased activity on AADA and AAAD, whereas the overall activity pattern for all muteins remains the same with all muteins converging towards PesCDA$^{nm}$ over time (Fig 8). To evaluate an indirect effect on subsite preference, we took a closer look at the D164N (Fig 7D). Although it is less active on all substrates compared to PesCDA$^{nm}$, its activity seems significantly higher on AAAD than on A4, whereas PesCDA$^{nm}$ seems to have similar activities on these 2 substrates. If D164N increases the preference for a GlcN unit at subsite +1, AADA is exacted to bind in binding mode [−1,+2] (AaDA) producing ADDA. Indeed, the opposite seems to be the case with D164N producing significantly less ADDA and more AADD compared to PesCDA$^{nm}$, leaving subsite +1 unoccupied. This shift in product pattern is strongest for D164N, although applying to all muteins of this position hinting at a decreased contribution to substrate binding if D164 is missing. These effects as well as the results from our MD simulations suggest that D164 is indeed important for substrate binding, although it is not absolutely required. Despite the increased activity of all muteins on A3D1 substrates, the overall activity pattern is very similar for all substrates, suggesting no preference for a GlcNAc or GlcN unit.

Results obtained for Y168 are inconclusive. Muteins with a strongly reduced activity only start to converge towards PesCDA$^{nm}$ after 48 h, whereas these muteins show significant activities on ADAA already after 2 h (Fig 8). Muteins such as Y168G and Y168S deacetylate ADAA even faster than A4 (Fig 7E). This effect is surprising at first, as Y168 is thought to establish a carbohydrate/aromatic interaction at subsite 0, but substitutions with smaller side chains are actually closer to the subsite −1 acetyl or amino group (Fig 1). Here, serine (in Y168S) could hydrogen bond with the GlcN amino group, which matches the increased activity of Y168S on ADAA, although other muteins such as Y168G, where similar interactions are not expected, also show an increased activity on ADAA. As Y168 is part of the rather flexible loop 4, which is mainly stabilized by Y168, additional residues might be involved here to different extents depending on the substituted residue. Still, the effects at play regarding the influence of Y168 on the subsite preferences at subsite −1 need further investigation.

One residue that could indirectly influence the −1 subsite preference of Y168 muteins is H199, whose location or protonation state might differ depending on the residue replacing Y168. Of all positions tested, H199 reveals the most interesting differences in their muteins'

activity pattern on ADAA and AADA compared to A4. Whereas some of the muteins discussed so far show significantly higher activities on some A3D1 substrates compared to A4, none of them reach activities on A3D1 similar to PesCDA$^{nm}$ deacetylating A4. This differs for certain muteins at position H199, reaching activities far exceeding these levels on ADAA and AADA and even reaching acetate release levels of PesCDA$^{nm}$ acting on A4 (Figs 7E and 8). H199, which is not protonated in our MD simulations, forms a hydrogen bond with the −1 sugar acetyl group in 1 out of 4 frames. Because H199 could be protonated in some molecules during our in vitro experiments, this hydrogen bond might actually be stronger. Taking a closer look at the acetylation pattern of A2D2 when AADA was used as a substrate, most muteins produce higher amounts of AADD compared to PesCDA$^{nm}$ placing the GlcN unit at subsite −1. The only exceptions are H199K and H199R, substituting the possibly positively charged histidine with positively charged residues. It appears that histidine, lysine, and arginine at this position result in a GlcNAc preference at subsite −1, leading to reduced activities on ADAA and AADA compared to A4. If any other residue is introduced at this position, this preference is removed, resulting in a higher acceptance of GlcN units, consequently increasing the normalized activity on ADAA and AADA compared to A4. Muteins such as H199Y and H199Q deacetylate ADAA as quickly as PesCDA$^{nm}$ deacetylates A4, indicating an even acceptance of GlcNAc and GlcN units. With their increased acceptance of GlcN units at subsite −1, these muteins are able to produce significant amounts of AADD or ADDD when deacetylating A4 and AADA or ADAA, respectively (AAaA → AADa → AADD; ADaA → ADDa → ADDD).

## Polymer *N*-acetylation

Sreekumar and colleagues showed that PesCDA produces larger GlcNAc blocks compared to a chemically *N*-acetylated control when polyglucosamine was *N*-acetylated enzymatically. This increased block size was attributed to a GlcNAc preference at subsite −1 [19]. Here, we were able to show that H199 is mainly responsible for this preference, which could be even stronger for H199K, whereas H199Y showed a more even acceptance of GlcNAc and GlcN, possibly even a preference for GlcN units. To evaluate whether or not this preference actually influences the GlcNAc block size, we *N*-acetylated polyglucosamine enzymatically with PesCDA$^{nm}$, H199Y, and H199K, taking chitosan samples with increasing $F_A$ (S8 Fig). Then, samples of comparable $F_A$ as well as chemically *N*-acetylated chitosans were digested with chitinosanase. This enzyme cleaves the substrate after the sequence DA, which is one unit after the transition from a D- to an A-block. Thus, the resulting products always contain a number of A- and D-units that equals the originating A- and D-block sizes. All digests were either separated by hydrophilic liquid interaction or size exclusion chromatography (HILIC or SEC) before being measured via MS or MS and refractive index (RI) detector, respectively. Based on the MS results of oligomeric products, the weight average A- and D-block size of the polymers was calculated (Fig 9A and 9B).

At similar $F_A$, chitosans produced with H199K have the largest A-blocks followed closely by those produced with PesCDA$^{nm}$, both are clearly larger compared to the chemical control and chitosans produced with H199Y. At first glance, H199Y produces A-blocks with a similar size compared to the chemical control. However, upon closer inspection, H199Y produces smaller A-blocks at higher $F_A$, which can be attributed to a larger proportion of A1D1 products found after chitinosanase digestion. This shift is even visible in the SEC chromatograms measured via RI detector (Fig 9C) where more dimeric products are visible for chitosans produced with H199Y. This hints at an alternating pattern produced by this PesCDA mutein. Besides these differences in oligomeric composition, the SEC chromatograms further highlight the strong

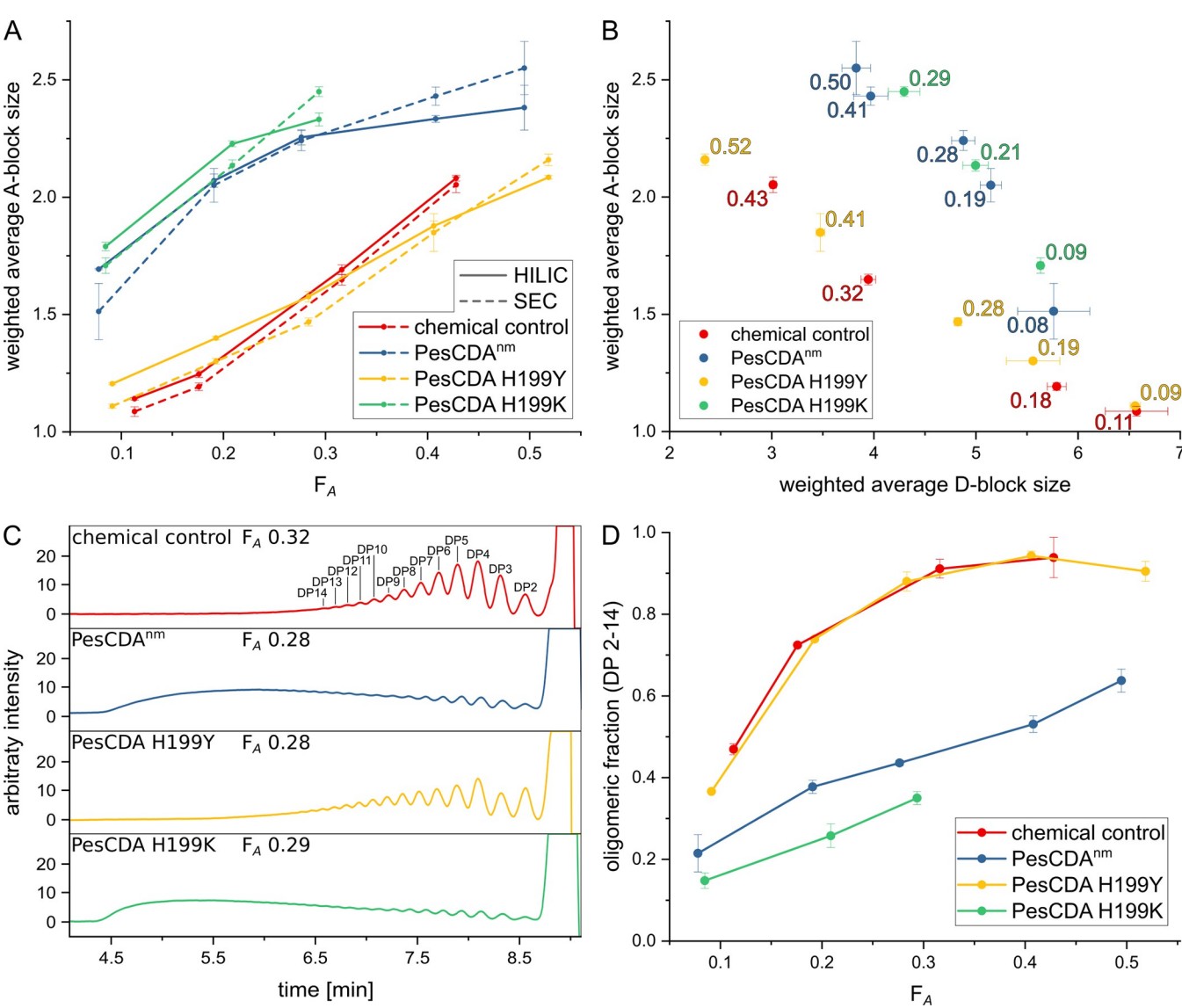

**Fig 9. Characteristics of chitosans produced via enzymatic and chemical *N*-acetylation.** (**A**) $F_A$-dependent weight average A-block size of chemically and enzymatically produced chitosan polymers. Fragments obtained after chitinosanase digestion were separated either via hydrophilic liquid interaction or size exclusion chromatography (HILIC or SEC) before mass spectrometry measurement. (**B**) Weight average A- and D-block sizes of chemically and enzymatically produced chitosan polymers. Fragments obtained after chitinosanase digestion were separated via SEC. For the corresponding HILIC results, see S9 Fig. (**C**) Representative SEC chromatograms of chitinosanase digests of different chitosans close to $F_A$ 0.3. Peaks are exemplary labeled for the chemical control. Exemplary SEC chromatograms for all $F_A$s can be found in S10 and S11 Figs. (**D**) $F_A$-dependent oligomeric fraction (peak area of all DP 2–14 peaks) of all fragments found in SEC chromatograms (*n* = 3). All values can be found in S5 Data.

differences between these chitosans. At similar $F_A$, both the chemical control and the chitosan produced with H199Y can be cleaved almost completely into DP 2 to 14 oligomers by chitinosanase if the $F_A$ reaches 0.3 (Fig 9C and 9D). For chitosans produced with PesCDA^{nm} and H199K, only 44% and 35% of all chitinosanase products reach a DP of 14 or lower, respectively. This clearly demonstrates that GlcNAc units in these chitosans are grouped into larger blocks, reducing the overall number of available A-blocks needed for latter chitinosanase cleavage. In addition, this further underlines the difference between PesCDA^{nm} and H199K, where the latter seems to produce even larger A-blocks.

## Discussion

Our results suggest that the catalytic mechanism of PesCDA resembles that of previously described CE4 enzymes, with D46 and H196 acting as the catalytic base and acid, respectively. D46 is stabilized by R137 and the $pK_a$ value of H196 is adjusted by D167. A divalent metal ion, bound by D47, H99, and H103, coordinates the substrate's acetyl group and the catalytic water, which acts as a nucleophile during catalysis. Some other residues studied here are likely involved in protein folding. These include R137, P139, I163, T65, D167, D197, and V198 with varying significances. Note that we mainly studied the interactions of the side chains, which were changed by mutation, whereas main chain interactions could only be observed through indirect effects. Therefore, other residues included in our SSM library could be important for protein folding, although not visible here.

Our main focus in this study was on the enzyme–substrate interactions not involved in catalysis, possibly causing subsite preferences for either GlcNAc or GlcN units. As previously shown, PesCDA converts A4 to AADA with high purity [4,48]. According to our data, this high specificity for binding mode [−2,+1] can be accounted for by mainly 3 or arguably 5 residues. First, N75 forms strong hydrogen bonds at subsites −2, −1, and, possibly, subsite −3, although the latter was only predicted by MD simulations. This makes N75 crucial for substrate binding, making it irreplaceable by any other amino acid without strongly or even completely loosing enzyme activity. Its interaction with the acetyl group at subsite −2 results in a preference for GlcNAc units at this subsite as evidenced by the reduced PesCDA activity on DAaA. Second, W76 shows carbohydrate/aromatic interactions with the subsite −2 sugar, further favoring the occupancy of this subsite. Similar to N75, its exchange mostly results in complete activity loss if not replaced by another aromatic amino acid. Third, D164 forms a hydrogen bond at subsite +1, reducing the binding energy if this subsite is occupied. If D164 is replaced by any amino acid that is not negatively charged, enzyme activity is drastically reduced, although not as strong as seen for N75 and W76. In sum, strong interactions at subsites −2 and +1 lead to their occupation being highly favorable, resulting in binding mode [−2, +1] of A4, DAAA, ADAA, and AAAD.

Because D164 is the only residue strongly binding the +1 sugar unit, it seems surprising that subsite +1 is still occupied in D164 muteins when deacetylating A4, so that only small amounts of AAAD are produced. We would have expected higher amounts of AAAD, in which case subsite −3 would have been occupied as well, giving access to an additional hydrogen bond formed with N75. As this is not the case, it follows that an additional effect is involved. Whereas the hydrophobic pocket, which is formed by F141 and L194, mainly accommodates the acetyl methyl group at subsite 0, it appears that solvent exposure of these residues lowers their contribution to substrate binding. This effect, also present in AngCDA MD simulations [32], seems to result in a higher occupancy of subsite +1 as well. Next to these residues, which are strongly involved in determining the binding mode, other residues also contribute to substrate binding. These include Y140 and Y168, both establishing carbohydrate/aromatic interactions with the subsite 0 sugar unit. In addition, the main chain nitrogen of Y140 forms a stable hydrogen bond with the acetyl group at subsite 0.

PesCDA does not only produce AADA in high purity, but it is suitable to produce this oligomer with a good yield as its activity on AADA is strongly reduced. This differs for other CE4 enzymes such as AnCDA, ArCE4A, BsPdaC, and AngCDA, which continue to deacetylate their substrate until solely the reducing end remains acetylated [8,32,49,50]. Previously published MD simulation results for AngCDA show that the binding energy for AAdA is strongly reduced, resulting in a high substrate fluctuation and the substrate even leaving the binding site in several simulations [32]. Although with a reduced binding energy, PesCDA binding

AAdA seems to be rather stable. Thus, we assume that PesCDA is able to bind its own A3D1 product AADA in an unproductive binding mode [−2,+1], whereas other binding modes seem unlikely, as previously proposed for ClCDA [51]. This can again be attributed to the residues mentioned above, interacting with A4 and also AADA at subsites −2 and +1; however, subsite preferences also influence the binding of deacetylated substrates such as AADA.

PesCDA$^{nm}$ activities on A4 and all A3D1 substrates show that a GlcN unit bound at subsites −2 and −1 (DAaA and ADaA) results in a reduced activity, whereas a GlcN unit at subsite +1 (AAaD) seems to yield an equal if not a higher activity compared to GlcNAc units bound at all 4 subsites (AAaA). Despite the varying activities on DAAA, ADAA, and AAAD, these substrates are still deacetylated in binding mode [−2,+1]. This differs for AADA, which has to bind in a different binding mode to be deacetylated. Based on the importance of subsite −2, we would have expected AADA to be converted to AADD; however, due to the apparent preference for a GlcNAc unit at subsite −1, two-thirds of AADA are converted to ADDA. At pH 7, even higher amounts of ADDA are produced as shown by Hembach and colleagues [48]. Under these conditions, the protonation rate of H199 and GlcN units is increased. Thus, the hydrogen bond between H199 and the −1 sugar unit acetyl group as well as the repulsion of H199 and positively charged GlcN units is stronger. At our experimental conditions with pH 8.5, H199K and H199R are still protonated, strongly favoring a GlcNAc unit at subsite −1, producing mainly ADDA (AaDA → ADDA). If H199 is replaced by any other not positively charged residue, more AADD is produced (AADa → AADD). This does not necessarily indicate a GlcN preference for these muteins at subsite −1 but may rather result from the strong contribution of subsite −2, which otherwise would not be occupied. However, H199Y and H199Q seem to at least accept GlcNAc and GlcN units equally well if not even preferring GlcN units at subsite −1. Our results suggest that H199 contributes most to the GlcNAc preference at subsite −1, although it is not the only residue responsible for this preference. W76Y may have an increased acceptance for a GlcN unit; however, based on our data, we are not able to safely draw this conclusion. The activity on ADaA seems to be increased as well if Y168 is substituted for some small amino acids, even though the underlying effect is not understood so far. Strikingly, PgtCDA, which was reported to produce AADD in good yields as seen here for PesCDA H199Y, contains a tyrosine in loop 1 (equivalent to W76Y) and loop 6 (equivalent to H199Y) and a small alanine in motif 4 (equivalent to Y168A) [47,48], almost matching the effects we have seen here for our PesCDA muteins. Only PesCDA Y168A is almost inactive in our screening, whereas other small and aliphatic substitutions seem to increase the acceptance for a GlcN unit.

We hypothesized before that subsite preferences strongly influence the PA CDAs generate on polymeric substrates. Here, we were able to show that mutations at a single position can alter the preference at subsite −1 in both directions (H199K: stronger GlcNAc preference than PesCDA$^{nm}$; H199Y: equal acceptance of GlcNAc and GlcN units). Accordingly, we expected these PesCDA muteins to produce significantly different PAs when *N*-acetylating polymers. Whereas the subsite preference can be adjusted at subsite −1, subsite −2 will keep its high GlcNAc specificity. Consequently, PesCDA with GlcNAc preferences at both minus subsites is expected to most likely *N*-acetylate a GlcN unit next to 2 GlcNAc units towards the reducing end, thus creating large A-blocks. This was indeed observed when *N*-acetylating polyglucosamine with PesCDA$^{nm}$, and even larger A-blocks were produced with H199K, matching its stronger preference for GlcNAc units at subsite −1. Unlike PesCDA$^{nm}$ and H199K, H199Y produced rather small A-blocks and our results even indicate an alternating pattern. This matches the subsite preferences observed during our experiments on oligomeric substrates. Subsite −2 preferably binds GlcNAc units, whereas subsite −1 accepts GlcNAc and GlcN to an equal extent. Because of a higher availability of GlcN units during *N*-acetylation, it is most likely that a GlcN unit gets *N*-

acetylated 2 units downstream of a GlcNAc units towards the reducing end, creating an alternating pattern of acetylation ($\cdots$ADdDD$\cdots \rightarrow \cdots$ADADd$\cdots \rightarrow \cdots$ADADA$\cdots$).

## Conclusion and outlook

In this study, we gained a detailed understanding of the PesCDA binding site. We were able to determine crucial residues for substrate binding and subsite preferences. For subsite −1, we could even show that a single amino acid substitution can change the subsite preference, influencing the products' PA on both oligomers and polymers.

In follow-up experiments, a combination of different mutations could further increase the preference for a GlcN unit at subsite −1, enabling the production of a stronger alternating pattern. However, no further changes to subsite preferences appear to be possible in PesCDA based on our results. The preference at subsite −2 is linked to an amino acid, which is pivotal for enzyme activity, whereas the acetyl group at subsite +1 does not interact significantly with the enzyme. Therefore, other CDAs are needed with different subsite preferences or where changes to subsite preferences are possible at subsites −2 and +1. A CDA with GlcNAc preferences at both minus subsites and subsite +1, for example, would be expected to produce even larger A-blocks compared to H199K. To generate more specific patterns such as alternating A- and D-blocks of a certain size, CDAs with more subsites are required, such as BmCDA8 or UmCda7 [26,52]. To gain a first insight into subsite preferences and to avoid the laborious preparation and screening of a SSM library, we showed that MD simulations can be a useful tool. Subsequently, site saturation mutagenesis studies can be performed for selected amino acids to evaluate whether subsite preferences can or cannot be altered. Clearly, there are numerous combinations of subsite preferences imaginable that would result in chitosans with novel PAs, especially for CDAs with elongated binding sites. Based on the requirements of certain applications, chitosans could then be tailored accordingly, for example, enhancing or hindering their biodegradability in a certain environment, possibly even giving rise to specific oligomer products upon digestions with a strongly sequence-dependent chitosan hydrolase.

## Supporting information

**S1 Fig. Screening results using different dilutions of CMBs.** The PesCDA^nm as well as an empty vector control were subjected to the fluorescamine-based screening. Using 3 glucosamine standards (15 μM, 50 μM, and 100 μM), the concentration of free primary amines were calculated. As expected, the empty plasmid control shows values close to 0, whereas PesCDA^nm shows an increasing amount of free primary amines formed with a decreasing dilution of CMBs. With a strong activity increase still being visible comparing the 1:10 to the 1:5 dilution, we settled on the 1:10 dilution assuming that the CMBs are saturated with enzymes. The results further show that with all dilutions less then 10% of the 500 μM A4 substrates were deacetylated. All values can be found in S1 Data.
(TIFF)

**S2 Fig. Comparison of the 27 SSM library residues from our SWISS-MODEL and an AlphaFold2 model.** The AlphaFold2 model from the AlphaFold Protein Structure Database [35,36] was aligned to our SWISS-MODEL with an RMSD = 1.341 Å on all atoms and an RMSD = 0.782 Å on the 27 SSM library amino acids. Our SWISS-MODEL is shown with blue, and the AlphaFold2 model is shown with red carbon atoms, respectively. For an easier overall comparison, the other colors and labels are reused from Fig 1.
(TIFF)

**S3 Fig. Heatmap with standard deviations from Fig 2.** Each column shows the standard deviation of all available muteins at a given position, as indicated below the matrix. Empty fields indicate missing muteins. The muteins are grouped by the properties—nonpolar, polar, charged (+/−), or aromatic—of the residue by which the wild-type amino acid was exchanged ($n = 3$–$4$). All values can be found in S1 Data.
(TIFF)

**S4 Fig. Heatmap with standard deviations from Fig 6.** Each column shows the standard deviation of all available muteins at a given position, as indicated below the matrix. Empty fields indicate missing muteins. The muteins are grouped by the properties—nonpolar, polar, charged (+/−), or aromatic—of the residue by which the wild-type amino acid was exchanged ($n = 4$). All values can be found in S3 Data.
(TIFF)

**S5 Fig. Heatmap with standard deviations from Fig 8.** Each column shows the standard deviation of all available muteins at a given position, as indicated below the matrix. Empty fields indicate missing or not tested muteins. The muteins are grouped by the properties—nonpolar, polar, charged (+/−), or aromatic—of the residue by which the wild-type amino acid was exchanged ($n = 4$). All values can be found in S3 Data.
(TIFF)

**S6 Fig. PesCDA muteins activity from positions N75, W76, and D164.** Each column shows the activity of all tested muteins at one position for the indicated incubation time and substrate. Missing or not tested muteins are grayed out. For orientation, the PesCDA$^{nm}$ is included in each column at the corresponding position of the wild-type (WT) amino acid. The muteins are grouped by their properties nonpolar, polar, charged (+/−), and aromatic. Corresponding standard deviations can be found in a separate heatmap in S7 Fig ($n = 4$). All values can be found in S3 Data.
(TIFF)

**S7 Fig. Heatmap with standard deviations from S6 Fig.** Each column shows the standard deviation of all available muteins at a given position, as indicated below the matrix. Empty fields indicate missing or not tested muteins. The muteins are grouped by the properties—nonpolar, polar, charged (+/−), or aromatic—of the residue by which the wild-type amino acid was exchanged ($n = 4$). All values can be found in S3 Data.
(TIFF)

**S8 Fig. *N*-acetylation of polyglucosamine with PesCDA$^{nm}$ and muteins.** $F_A$ determined over time for *N*-acetylation of polyglucosamine using PesCDA$^{nm}$, PesCDA H199Y, and PesCDA H199K. Samples analyzed here with $F_A$'s close to 0.1, 0.2, 0.3, 0.4, and 0.5 were used for chitinosanase digestion to determine the PA of these chitosans ($n = 1$). All values can be found in S5 Data.
(TIFF)

**S9 Fig. Weight average A- and D-block size of chemically and enzymatically produced chitosan polymers.** Obtained fragments after chitinosanase digestion were separated via hydrophilic liquid interaction chromatography (HILIC). For the corresponding figure measured via size exclusion chromatography, see Fig 9B ($n = 3$). All values can be found in S5 Data.
(TIFF)

**S10 Fig. Size exclusion chromatograms of chitinosanase digested chitosans (chem. control and PesCDA$^{nm}$).** Samples were measured in triplicates. Here, an exemplary chromatogram is

shown which best resembles all replicates.
(TIFF)

**S11 Fig. Size exclusion chromatograms of chitinosanase digests of different chitosans (PesCDA H199Y and PesCDA H199K).** Samples were measured in triplicates. Here, an exemplary chromatogram is shown which best resembles all replicates.
(TIFF)

**S12 Fig. Nucleotide and amino acid sequence of NSt-MBP-PesCDA-VcCBD1+2-CSt.** All features are annotated in the green underlying box.
(TIFF)

**S1 Data. Spreadsheet with all results from the first screening of all 27 positions without and with outlier elimination.** An additional sheet contains the results from the initial test screen with different dilutions of CMBs (S1 Fig).
(XLSX)

**S2 Data. Spreadsheet with all results from the MD simulations.** This spreadsheet contains the results for all substrates and binding modes divided into separate sheets within one excel document. Each sheet contains the RMSF, hydrogen bond occupancy, and BFE results for all replicates of this substrate/binding mode combination. Further information can be found in the spreadsheet itself.
(XLSX)

**S3 Data. Spreadsheet with all MS[1] results from the detailed screening.** The spreadsheet contains several sheets, in which the first two give the raw data for all replicates and their averages and standard deviations. The following sheets each give either the relative acetate release (rar) for the given substrate or the rar normalized to PesCDA[nm], which is shown in most of the figures in this article. Further information can be found in the spreadsheet itself.
(XLSX)

**S4 Data. Spreadsheet with all MS[2] results from the detailed screening.** The spreadsheet contains 2 sheets, in which the first one gives the raw data for all replicates and their averages and standard deviations. The second sheet contains only the average values formatted to allow an easier comparison of the data between the different muteins and the PesCDA[nm]. Further information can be found in the spreadsheet itself.
(XLSX)

**S5 Data. Spreadsheet with all results from the polymer analysis.** The spreadsheet contains 4 sheets. The first sheet contains the $F_A$ determination, which is plotted in S8 Fig. The second and third sheet give the weight average A- and D-block sizes determined via HILIC-MS (see S9 Fig) and via SEC-MS (see Fig 9B), respectively. The last sheet contains the areas for the oligomer and polymer fractions from the SEC chromatograms (see S10 and S11 Figs) that were used to calculated the oligomeric fractions shown in Fig 9D.
(XLSX)

## Author Contributions

**Conceptualization:** Martin Bonin, Stefan Cord-Landwehr, Antoni Planas, Bruno M. Moerschbacher.

**Data curation:** Martin Bonin.

**Formal analysis:** Martin Bonin, Antonia L. Irion.

**Funding acquisition:** Antoni Planas, Bruno M. Moerschbacher.

**Investigation:** Martin Bonin, Antonia L. Irion, Anika Jürß.

**Methodology:** Martin Bonin, Antonia L. Irion, Anika Jürß, Sergi Pascual.

**Project administration:** Stefan Cord-Landwehr, Antoni Planas, Bruno M. Moerschbacher.

**Resources:** Antoni Planas, Bruno M. Moerschbacher.

**Software:** Martin Bonin.

**Supervision:** Sergi Pascual, Stefan Cord-Landwehr, Antoni Planas, Bruno M. Moerschbacher.

**Validation:** Martin Bonin, Antonia L. Irion, Stefan Cord-Landwehr, Bruno M. Moerschbacher.

**Visualization:** Martin Bonin, Anika Jürß.

**Writing – original draft:** Martin Bonin.

**Writing – review & editing:** Martin Bonin, Anika Jürß, Stefan Cord-Landwehr, Antoni Planas, Bruno M. Moerschbacher.

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
