## [Editor Report · Decision Letter 0]

24 Mar 2023

Dear Dr Moerschbacher, 

Thank you for submitting your manuscript entitled "Knowledge based mutation of crucial residues in Pestalotiopsis sp. chitin deacetylase to generate tailor made chitosan polymers" for consideration as a Research Article by PLOS Biology. Please accept my apologies for the delay in getting back to you as we consulted with an academic editor about your submission.

Your manuscript has now been evaluated by the PLOS Biology editorial staff, as well as by an academic editor with relevant expertise, and I am writing to let you know that we would like to send your submission out for external peer review.

Before sending your manuscript to reviewers, we would like to suggest the following modification to the title:

"Knowledge based mutation of crucial residues in a chitin deacetylase to generate tailor-made chitosan polymers"

In addition, we need you to complete your submission by providing the metadata that is required for full assessment. To this end, please login to Editorial Manager where you will find the paper in the 'Submissions Needing Revisions' folder on your homepage. Please click 'Revise Submission' from the Action Links and complete all additional questions in the submission questionnaire.

Once your full submission is complete, your paper will undergo a series of checks in preparation for peer review. After your manuscript has passed the checks it will be sent out for review. To provide the metadata for your submission, please Login to Editorial Manager (https://www.editorialmanager.com/pbiology) within two working days, i.e. by Mar 26 2023 11:59PM.

Kind regards,

Richard

Richard Hodge, PhD

Associate Editor, PLOS Biology

rhodge@plos.org

PLOS

---

## [Decision Letter · Decision Letter 1]

5 May 2023

Dear Dr Moerschbacher,

Thank you for your patience while your manuscript "Knowledge based mutation of crucial residues in a chitin deacetylase to generate tailor-made chitosan polymers" was peer-reviewed at PLOS Biology. Please accept my apologies for the delays that you have experienced during the peer review process. Your manuscript has now been evaluated by the PLOS Biology editors, an Academic Editor with relevant expertise, and by three independent reviewers. 

In light of the reviews, which you will find at the end of this email, we would like to invite you to revise the work to thoroughly address the reviewers' reports.

As you will see below, the reviewers are generally positive about your manuscript and think the findings are interesting. However, they raise some overlapping concerns, including the qualitative nature of the analyses to measure enzyme activity and ask that direct quantitative assessments of enzyme activity are conducted in the revision.

Given the extent of revision needed, we cannot make a decision about publication until we have seen the revised manuscript and your response to the reviewers' comments. Your revised manuscript is likely to be sent for further evaluation by all or a subset of the reviewers.

**IMPORTANT - SUBMITTING YOUR REVISION**

*Re-submission Checklist*

*Published Peer Review*

*PLOS Data Policy*

*Blot and Gel Data Policy*

Sincerely,

Richard

Richard Hodge, PhD

Associate Editor, PLOS Biology

rhodge@plos.org

REVIEWS:

Reviewer #1: This manuscript by Bonin et al presents a study on the amino acid residues that comprise the substrate binding subsites of a chitin deacetylase to inform the generation of tailor made variants with altered properties. A combination of protein engineering and theoretical considerations involving extensive computer modelling was used to analyze the chitin deacetylase from Pestalotoipsis sp. The study identified several amino acids that would appear to confer substrate specificity and this was used to engineer enzyme variants with apparent enhanced activity toward specific substrates. The manuscript is organized logically and written clearly - it is concise while presenting all pertinent information. The interpretation of the data based on homology modeling, substrate docking and molecular dynamic simulations in combination with the generation and analysis of 513 muteins (470 verified by sequencing) is logical. That said, a major issue that the authors need to address is their use of (more or less) qualitative data regarding the "activity" of the muteins produced. The activity levels observed and subsequently discussed do not appear to be specific activities, which would account for the concentration of enzyme present. The specific activity of a mutein would directly report on the effect of an amino acid replacement, and thus identify critical residues, in this case for substrate binding. However, the activities reported do not account for enzyme concentration and the assumption is that each mutein studied is produced at the same level. Of course, with 470 muteins produced, determination of specific activities would require significant effort, but some sense of enzyme production would be needed to support the interpretations made as currently stated. Without these analyses, the authors need to acknowledge this shortfall with their data, and discuss it accordingly.

Reviewer #2: This manuscript by Bonin et al. presents an extensive site-directed mutagenesis campaign to alter the specificity of a chitin deacetylase toward individual N-acetyl groups on chitin oligosaccharides. The results represent a large amount of work, including the production and assay of site-saturation mutants of over two-dozen amino acid positions, followed by further refined study. The manuscript is very well produced regarding writing and data representation and the conclusions are well-supported.

However, I do not recommend this manuscript for publication in PLoS Biology. Fundamentally, this is a classical protein engineering study, which would be more appropriate for a field-specific biochemical or biotechnology journal. From the perspective of PLoS Biology, this study really provides no fundamental insight into biological processes. Conceptually and methodologically, the study generally uses well-worn protein engineering and analysis approaches. Likewise, the selection, engineering and application of chitin deacetylases to produce specific oligosaccharide acetylation variants has also been explored, notably by this group (e.g. refs. 20, 27, 30) and others. Notwithstanding the comments in the first paragraph above, it does not seem that this study meets the bar of "works of exceptional significance, originality, and relevance in all areas of biological science".

Reviewer #3: In their article Bonin et al. use site saturation mutagenesis to investigate the role of active site amino acid residues play in catalysis for the Pestalotiopsis sp. chitin deacetylase. These muteins were investigated through screening with a fluorogenic detection reagent, and through mass spectroscopy analysis of released products. The authors give some interesting insight into the role of H199 muteins in catalysis and characterize these mutants in both N-deacetylation and N-acetylation assays.

Although the authors do show differences in activity for various mutants, the conclusions that they draw from these experiments are confounded by the lack of enzyme quantification performed. The decrease (or increase) in any activity observed in the many assays performed in this work may be either a result of genuine changes in the catalytic rate or a change in purification yield – that may be due to stability changes, codon usage or protein folding rates. The conclusions drawn from the screening experiments are therefore difficult to support, and their discussion should be curtailed.

Major issues:

p. 7, L. 140 The arbitrary choice of outliers makes the quality of the data dubious. There are statistical tests that can be performed to identify outliers, and these should be used in place of the automated ‘outlier’ removal criteria.

P5, L80 The authors should justify why swiss-model was used instead of Alphafold2. They should also state whether any of the amino acids investigated here have large changes in positioning in the alfafold2 model.

p. 12, L. 286 Introducing the CBDs from Vibrio cholerae CDA results in a chimera protein. Have the authors investigated the effect of these CBDs on enzyme activity? The CBD and the catalytic domains may interact, therefore the effects of introducing these domains from one homolog into another should be quantified and reported. Furthermore, as this is a chimeric protein it is inappropriate to refer to this protein as the ‘Wild-type’.

p. 13, L. 295 Using chitin affinity to ensure similar enzyme concentrations while using a CBD would be fine if you weren't also modulating the enzyme's affinity for chitin by mutating the residues that can modify the chitin on purification beads. Therefore, it appears improbable this purification method results in a similar interaction pattern with the beads. To quantify differences in enzymatic activity for the muteins you should quantify the purified enzyme present in you experiments. Especially since you mentioned the possibility of low expression yields being a concern (p. 14, L. 304). This might occur due to folding problems being introduced from mutating the primary sequence of the protein, as you also note at p. 16, L. 349, or it may simply be due to one protein expressing better than another.

p. 14, Fig 2. These images are not easy to interpret due to the font being very small. Consider moving the numbers to the SI, combining with the SD and p-value for a clearer overview. It is also not clear which results (if any) are significantly different from the unmutated protein. Without the significance of the results it is difficult to make any meaningful conclusions from this data.

p. 15, L. 323 This explanation does not account for the possibility that this mutant facilitates a different binding mode. To claim it is a false positive you should do more experiments to test this assumption. For example, re streak the clone, verify the plasmid sequence, quantify the protein being purified and test the activity again.

p. 24, L. 536 This sentence refers to an image that indeed shows more products being formed after 48h. There is a lack of explanation to later refer to this as a discussion on it (p. 26, L. 625). Why does this happen? Could it be explained by WT product inhibition or re-acetylation of GlcN being more pronounced than it is for these muteins?

P. 18 Fig.3 It is inappropriate to report SD values with so many significant digits as it gives a false impression that you know how much SD there is. Values should be only be reported with significance limited to the SD. For example. Your value of -27.01 +/- 11.70 should be reported as ‘-30 +/- 10’ . Also, your value of 0.05 +/- 0.00 gives the false sense that there is no SD, you should report this as 0.050 +/- 0.002 or whatever values you have calculated for this.

P. 17/18. Please include a note on the correspondence between the docking calculation methods that you have used and their experimental calculations of binding energy. This will help the reader evaluate the significance of the values determined here.

P.20 The results of the docking experiments are discussed as though they are experimentaly determined interactions and energies. For example, on line 460 : ‘ the side chain nitrogen and oxygen form hydrogen bonds with the -1 sugar in binding mode …’. This statement has however not been experimentally validated; this is a hypothesis.

P. 20 The term ‘stack’ is repeatedly used to refer to interactions between aromatic acid amino acids and sugar residues. This term is ambiguous, and it makes the reader think of pi-pi stacking interactions, which clearly cannot be present with the sugar residue. If the authours intend to state that there are supposed CH-pi interactions, they should calculate whether these are within the proper distances and at the proper angles to be true interactions.

Minor Issues:

p. 6 , L. 105 Why did you choose to test the activity on an A5 substrate when the rest of the research appears to have been focused on the A4 substrate?

p. 10, L. 204 Discuss here why are only these 2 mutants were used in the experiment, based on what metric were these selected? Mentioning this in the last part of your results and discussion is to late.

p. 12, L. 272 Following the A = GlcNAc and D = GlcN labeling and your explanation that A4 is a GlcNAc tetramer, the D46 could refer to a 46-mer of GlcN instead of aspartic acid 46. Updating the carbohydrate abbreviations is advised to avoid confusing amino acids and sugars.

p. 13, Fig 1. Colour-coding the signal peptide and substrate both green is inconsistent and confusing. Could be improved by adding another colour for the signaling peptide.

p. 22, L. 498 You should specify the 'type of sugar' to the acetylation state of the glucosamine. Written like this you suggest that any sugar goes.

p. 27, L. 631 Could this be accounted for by protein folding deviations when the hydrophobic pocked isn't formed?

p. 41, figure Needs replicates.

p12, L. 268 consider the term docked instead of ‘placed.’

p24. L 543 the term ‘catchup’ is ambiguous, consider using better terminology.

P23. Figure 6. These charts would be more interpretable if expressed as rates, either as rates of substrate decay or as initial rates where appropriate.

P 29 L 683. Presence of a positively charged amino acid H199 lead to charge-charge repulsion with a deacetylated glucosamine residue? Is this why you the specificity pattern

---

## [Decision Letter · Decision Letter 2]

9 Nov 2023

Dear Bruno,

Thank you for your patience while we considered your revised manuscript "Knowledge based mutation of crucial residues in a chitin deacetylase to generate tailor-made chitosan polymers" for publication as a Research Article at PLOS Biology. This revised version of your manuscript has been evaluated by the PLOS Biology editors, the Academic Editor and two of the original reviewers.

Based on the reviews, I am pleased to say that we are likely to accept this manuscript for publication, provided you satisfactorily address the remaining points raised by the Reviewer #3. In addition, I would be grateful if you could please also make sure to address the following data and other policy-related requests that I have provided below (A-D):

(A) We would like to suggest the following modification to the title: 

“"Engineering of a chitin deacetylase to generate tailor-made chitosan polymers"

(B) You may be aware of the PLOS Data Policy, which requires that all data be made available without restriction: http://journals.plos.org/plosbiology/s/data-availability. For more information, please also see this editorial: http://dx.doi.org/10.1371/journal.pbio.1001797

-Supplementary files (e.g., excel). Please ensure that all data files are uploaded as 'Supporting Information' and are invariably referred to (in the manuscript, figure legends, and the Description field when uploading your files) using the following format verbatim: S1 Data, S2 Data, etc. Multiple panels of a single or even several figures can be included as multiple sheets in one excel file that is saved using exactly the following convention: S1_Data.xlsx (using an underscore).

-Deposition in a publicly available repository. Please also provide the accession code or a reviewer link so that we may view your data before publication. 

Figure 2, 3, 4A-B, 6, 7, 8, 9A-D, S1, S3, S4, S5, S6, S7, S8, S9

(C) Please also ensure that each of the relevant figure legends in your manuscript include information on *WHERE THE UNDERLYING DATA CAN BE FOUND*, and ensure your supplemental data file/s has a legend.

(D) Please ensure that your Data Statement in the submission system accurately describes where your data can be found and is in final format, as it will be published as written there. 

We expect to receive your revised manuscript within two weeks. 

*Published Peer Review History*

*Press*

Kind regards,

Richard

Richard Hodge, PhD

rhodge@plos.org

Reviewer remarks:

Reviewer #1: My issue has been adequately addressed.

Reviewer #3: Regarding the WT nomenclature of PesCDA. The construct described in the paper is not a wild-type protein, it is a fusion construct. Naming it as a wild-type protein, no matter what the reasoning behind it is factually incorrect. It is not a wild-type protein. 

A method to determine whether CH/pi interactions are within reasonable distances and angles can be found in https://doi.org/10.1021/jacs.5b08424 by Hudson et. al. See Figure 3. The author should apply these calculations to CH-pi interactions that they 'visually evaluate'.

---

## [Editor Report · Decision Letter 3]

4 Dec 2023

Dear Bruno,

On behalf of my colleagues and the Academic Editor, Chaitan Khosla, I am pleased to say that we can accept your manuscript for publication, provided you address any remaining formatting and reporting issues. These will be detailed in an email you should receive within 2-3 business days from our colleagues in the journal operations team; no action is required from you until then. Please note that we will not be able to formally accept your manuscript and schedule it for publication until you have completed any requested changes.

PRESS

Best wishes, 

Richard

Richard Hodge, PhD

rhodge@plos.org

PLOS
